# Improving the Strength of Sandy Soils via Ureolytic CaCO$_3$ Solidification by *Sporosarcina ureae*

Justin Michael Whitaker[1], Sai Vanapalli[2], and Danielle Fortin[1];

[1]Department of Earth and Environmental Sciences (413-ARC). University of Ottawa, K1N 6N5, Ottawa, ON, Canada
[2]Department of Civil Engineering (A015-CBY). University of Ottawa, K1N 6N5, Ottawa, ON, Canada

*Correspondence to:* D. Fortin (dfortin@uottawa.ca)

Key words: Urease, calcite precipitation, MICP, *Sporosarcina, Bacillus,* biomineralisation, biofilm

**Abstract**

'Microbial induced carbonate precipitation' (MICP) is a biogeochemical process that can be applied to strengthen materials. The hydrolysis of urea by microbial catalysis to form carbonate is a commonly studied example of MICP. In this study, *Sporosarcina ureae*, a ureolytic organism, was compared to other ureolytic and non-ureolytic organisms of *Bacillus* and *Sporosarcina* in the assessment of its ability to produce carbonates by ureolytic MICP for ground reinforcement. It was found that *S. ureae* grew optimally in alkaline (pH ~9.0) conditions which favoured MICP and could degrade urea (units [U] /mL = μmol/min.mL.OD$_{600}$) at levels (30.28 U/mL) similar to *S. pasteurii* (32.76 U/mL), the model ureolytic MICP organism. When cells of S. ureae were concentrated (OD$_{600}$ ~15-20) and mixed with cementation medium containing 0.5 M calcium chloride (CaCl$_2$) and urea into a model sand, repeated treatments (3 x 24 h) were able to improve the confined direct shear strength of samples from 15.77 kPa to as much as 135.80 kPa. This was more than any other organism observed in the study. Imaging of the reinforced samples with scanning electron microscopy and energy dispersive spectroscopy confirmed the successful precipitation of calcium carbonate (CaCO$_3$), across sand particles by *S. ureae*. Treated samples were also tested experimentally according to model North American climatic conditions to understand the environmental durability of MICP. No statistically significant (p < 0.05, n = 3) difference in strength was observed for samples that underwent freeze-thaw cycling or flood-like simulations. However, shear strength of samples following acid-rain simulations fell to 29.2% of control MICP samples. Overall, the species *S. ureae* was found to be an excellent organism for MICP by ureolysis to achieve ground strengthening. However, the feasibility of MICP as a durable reinforcement technique is limited by specific climate conditions (i.e. acid rain).

# 1 Introduction

Biomediated calcium carbonate ($CaCO_3$) production is the process by which organisms induce the precipitation of calcium carbonate. With reference to bacterial $CaCO_3$ precipitation, also known as, 'microbial induced carbonate precipitation', 'microbial induced calcite precipitation' (MICP) and 'microbial induced calcium carbonate precipitation' (MICCP), the phenomenon is well documented (Stocks-Fischer et al., 1999; Dejong et al., 2006; Whiffin et al., 2007; van Paassen et al., 2010). For example, cyanobacteria precipitate $CaCO_3$ in microbial processes related to the shedding of the S-layer, forming the stalagmites and stalactites in limestone caves and adding to the rocky sediments of coral reefs (Southam 2000). Crystal aggregation of $CaCO_3$ in the kidney, urinary tract or gallbladder have been shown to be induced by microorganisms such as *Proteus mirabilis*, a urease positive organism due to secondary infection (Worcester and Coe 2008). Ureolytic soil organisms of the species *Sporosarcina* or *Bacillus*, can also induce $CaCO_3$. For example, in their cycling of nitrogen with a urease enzyme (Hammes et al., 2003; Gower 2008; Worcester and Coe 2008). This last group of MICP producers has peeked recent engineering interests to apply them in a bioengineering and repair context.

MICP biotechnology utilizing ureolytic soil organisms, most notably *Sporosarcina pasteurii*, has been shown to directly reinforce or restore engineered or natural structures, such as the repair of historical monuments (Le Métayer-Levrela et al., 1999; Webster and May 2006), marble slabs (Li and Qu 2011) and stone heritage sites (Rodriquez-Navaro et al., 2012) and reduce weathering of soil embankments (Chu et al., 2012). The enzyme urease (urea amidohydrolase, E.C. 3.5.1.5) initiates the process, catalyzing the breakdown of urea to raise local pH and produce $CaCO_3$ in a solution of calcium ions often supplied as calcium chloride ($CaCl_2$), as summarized in equations 1 and 2 (eq. [1, 2]). The produced $CaCO_3$ fills structural gaps or bridges materials (i.e., soils grains, etc.) to form a cemented product with unconfined strengths of up to 20 MPa (Whiffin et al., 2007).

$$[1]\ CO(NH_2)_2 + 2\ H_2O <--> 2\ NH_4^+ + CO_3^{2-}\ (\text{Urea Hydrolysis})$$
$$[2]\ 2\ NH_4^+ + CO_3^{2-} + CaCl_2 <--> CaCO_3 + 2\ NH_4Cl\ (CaCO_3\ \text{Formation})$$

Bacterial species such as *Bacillus sphaericus* (van Tittelboom et al., 2010) and *Bacillus megaterium* (Krishnapriya et al., 2015) have also been applied in material or volume strengthening. The aforementioned ureolytic soil organisms are attractive for MICP as they are, 'generally regarded as safe', (GRAS) bacteria with accessible substrates (i.e., urea) and an aerobic metabolism applicable to most engineering and terrestrial environments (DeJong et al., 2006). These gram positive organisms offer other attractive features such as spore forming capability allowing for long term capsule storage in cements (Jonkers 2011) and exopolysaccharide (EPS) secretion for improved material bonding (Bergdale 2012).

The application of MICP in industry as a biotechnology is proposed to help reduce the need for current structure repair practices such as chemical grouting, which have been found to be environmentally detrimental in its permanence (DeJong et al., 2010) and, in some cases, posing serious human health risks (Karol 2003). That said, ureolytic MICP does produce excess ammonia which can be harmful (van Paassen et al., 2010). The use of

nitrifying and denitrifying bacteria could help solve this issue by oxidizing ammonia to nitrate and later nitrogen gas without affecting MICP. In fact, the work of Gat et al. (2014) has shown co-cultures of ureolytic and non-ureolytic bacteria can actually be beneficial to MICP. Alternatively, denitrifying bacteria can be used to directly induce MICP to avoid ammonia toxicity, though the level of $CaCO_3$ is comparatively less to ureolytic MICP and harmful nitrites can build up in solution (van Paassen et al. 2010). Other pathways to achieve MICP have also been explored with *B. megaterium* and *B. sphaericus* (Kang et al., 2015; Li et al., 2015).

Problems on large scale application of the MICP technology have occurred too and remain unsolved. Research by van Paassen et al. (2009) found poor sample homogeneity of MICP as well as decreasing biomass and urease-inducing $CaCO_3$ activity over time and increasing soil depth in a pilot 100 m$^3$ sand study using *Sporosarcina pasteurii*, attributing these heterogeneities mostly to the application process. Alternative metabolisms and bacteria for large scale applications in biomineralization of $CaCO_3$ have also been investigated by the group (van Paassen et al., 2010). Indeed, it has been commented that the type of bacteria utilized is one of the major considerations and potential limitations in large scale geotechnical operations (Mitchell and Santamarina, 2005).

Therefore, the search for new bacteria by which to achieve viable levels of MICP is important for optimizing the protocol best suited (in terms of performance, economics and environmental impact) for marketing in green industry (van Paassen et al., 2010; Cheng and Cord-Ruwisch 2012; Patel 2015). Following a literature review of the nine documented species of *Sporosarcina* (Claus and Fahmy, 1986), seven species were found to be urease positive and distinct from *Sporosarcina pasteurii* as alternative ureolytic MICP sources. While no candidate improves on some of the short comings of ureolytic MICP (i.e., ammonia toxicity), each candidate was found to be poorly investigated in the current MICP technology, despite fitting the ureolytic model for MICP. One candidate, *Sporosarcina ureae* was selected at random for investigation as it was deemed appropriate to explore the feasibility of a single candidate species in thorough comparison to other, already published species applied in ureolytic MICP.

Thus, the primary goal of this study was to investigate the suitability of *S. ureae* as a MICP organism in material improvement by testing it experimentally against the previously investigated species of *Sporosarcina pasteurii*, *Bacillus megaterium* and *Bacillus sphaericus*. In its assessment, a parallel investigation was also performed to assess how the MICP technology, utilizing *S. ureae* as the candidate MICP organism, can perform under various environmental conditions including acid rain, flooding and freeze-thaw cycling concurrent with colder North American climates.

**2 Materials and methods**

**2.1 Bacteria strains, media, culture and stock conditions**

Strains of *Sporosarcina ureae* (BGSC 70A1), *Bacillus megaterium* (BGSC 7A16), *Lysinibacillus sphaericus* (BGSC 13A4; previously known as *Bacillus sphaericus* [Ahmed et al., 2007]) and *Bacillus subtilis* (BGSC 3A1$^T$) were obtained from the Bacillus Genetic Stock Centre (BGSC).

*Sporosarcina pasteurii* (ATCC 11859), previously known as *Bacillus pasteurii* (Yoon et al., 2001), was kindly donated by the group of Rodrigues *et al.* (University of Houston, USA). *Escherichia coli* DH5a$^{TM}$ was obtained from ThermoFisher. *S. ureae* and *S. pasteurii* strains were grown at 30 $^o$C in a modified ATCC 1832 medium as follows: 5 g/L yeast extract (YE)

(BD Bacto[TM]), Tris-Base (Trizma[TM]), 5 g/L ammonium sulfate (Molecular biology grade, Sigma-Aldrich), 10 g/L urea (Molecular biology grade, Sigma-Aldrich), pH 8.6 . The culture broth, ATCC Medium 3 (3 g/L Beef extract [BD Bacto[TM]]

and 5 g/L peptone [BD Bacto[TM]]) was used for *B. megaterium*, *L. sphaericus* and *B. subtilis*. and grown at 30 $^{o}$C, unless otherwise specified. Colonies of *Bacillus* and *Sporosarcina* were maintained on plates prepared as described supplemented with 15 g/L agar [BD Difco[TM]] . *E. coli* was grown in Luria-Bertani (LB) broth (10 g/L tryptone [Molecular biology grade, Sigma-Aldrich], 5 g/L yeast extract [BD Bacto[TM]], 10 g/L NaCl [Molecular biology grade, Sigma-Aldrich], pH 7.5) and maintained on LB plates at 37 $^{o}$C supplemented with 15 g/L agar (BD Difco[TM]). Long term stocks of all cultures were

prepared as described (Moore and Rene, 1975) but using dry ice as the freezing agent.

### 2.2 Chemical and Biological Analysis

### 2.2.1 Culturing


Single colonies were lifted and grown overnight at 200 RPM in 5mL of respective strain culture medium in a 15 mL Corning Falcon[©] tube. The overnight stock was combined with 200 mL of appropriate culture medium in a 500 mL Erlenmeyer flask and cultured at 175 RPM. The optical density at 600 nm ($OD_{600}$) was used to track changes in turbidity of a culture volume using a Biomate UV-Vis spectrophotometer (Thermoscientific) where 1 mL of culture volume was placed

into 1.5mL polystyrene cuvettes (BioRad) of a 1 cm path length. Ultra-pure water (ddH$_2$O) was used as a blank. At $OD_{600}$ values greater than 0.4, samples of culture volumes were diluted 10-100X in Tris buffered saline (TBS; 50 mM Tris-base [Trizma[©], Sigma-Aldrich], 150 mM NaCl [Molecular biology grade, Sigma-Aldrich], pH 7.5) to maintain a linear relationship between turbidity and cell growth. When $OD_{600}$ reached ~ 0.5, the culture was twice spun at 5000 RPM for 5 minutes followed by a pellet re-suspension in 50 mL TBS each time. Next a fraction of volume was removed, spun at 5000

RPM for 5 minutes and re-suspended ($OD_{600}$ ~ 0.2) in 200 mL of a urea broth (UB) medium in a 500 mL Corning PYREX[©] round glass media storage bottle containing a modified Stuart's Broth (Stuart et al., 1945) as follows: 20 g/L Urea (BioReagent, Sigma-Aldrich), 5 g/L Tris-Base (Trizma[©], Sigma-Aldrich), 1 g/L glucose (Reagent grade, Sigma-Aldrich), pH 8.0, with (UB-1) or without (UB-2) 10 g/L yeast extract (YE)  (BD Difco[TM]).  A negative control included a medium only condition. All steps were performed aseptically with preparations incubated at 150 RPM at 30 °C in triplicate for each

medium condition: UB-1 and UB-2.  Each culture for a medium condition was staggered 10 min apart and observed for 12 h, with duplicate 2.5 mL aliquots aseptically withdrawn every 1hr, beginning at time zero (t = 0 h). The entire protocol was performed twice for a total of 6 data sets (n = 6), measured in duplicate, per culture in a single medium condition.

### 2.2.2 Total Ammonia (NH$_3$-NH$_4^+$), pH and growth (OD$_{600}$) aliquots


To evaluate different cell parameters efficiently, duplicate aliquots (2.5mL) were taken for tracking pH, $OD_{600}$ and NH$_3$-NH$_4^+$ production. In brief, first, whole aliquot volume pH was taken with a SB20 symphony pH probe (VWR). Next, a 1mL volume was removed for $OD_{600}$ reading as described (2.2.1). Finally, a 500 uL sample for NH$_3$-NH$_4^+$ analysis was retrieved and diluted in 500 uL of ddH$_2$O and stored as described by HACH Inc. (Hach Co. 2015) with the following additional

modifications made: -20 $^{o}$C storage, 1 drop 5 N H$_2$SO$_4$.

### 2.2.3 Spectrophotometric analysis of $NH_3$-$NH_4^+$

Samples were thawed and neutralized with 5 N NaOH as described by HACH Inc. (Hach Co. 2015). $NH_3$-
$NH_4^+$ measurements were then performed as outlined (HACH Co. 2015) based on an adaptation of the work by Reardon et al. (1966) using a portable DR2700 HACH spectrophotometer. Samples were brought to a measureable range (0.01 to 0.50 mg/L $NH_3$-N) where required. Measurements for appropriate dilutions were made by mass and corrected to volume assuming a density of 1 g/L. Final values were reported as, 'U/mL' where units U = µmol of $NH_3$-$NH_4^+$ produced per minute and mL = mL solution normalized to culture density ($OD_{600}$) starting from t = 1 h.


### 2.3 Microbial cementation

### 2.3.1 Model sand

Industrial quality, pure coarse silica sand (Unimin Canada Limited) was examined with the following grain distribution where $D_{10}$, $D_{50}$, $D_{60}$ are 10 %, 50 % and 60 % of the cumulative mass: $D_{10}$ = 0.62 mm, $D_{50}$ = 0.88 mm, $D_{60}$ = 0.96 mm. The uniformity coefficient, $C_u$ was 1.55 indicating a poorly graded (i.e. uniform) sand as designated by the Unified Soil Classification System (USCS) (ASTM, 2017). A poorly graded soil was used as a model due to its undesirable geotechnical characteristics in construction (i.e., settling) and tendency for instability in nature (i.e.,
liquefaction) (Nakata et al., 2001; Scott, 1991).

### 2.3.2 Cementation medium (CM) and culture

Cells of each strain were grown in 1L of their respective medium split into two 1 L Erlenmeyer flasks containing
500 mL medium each at 175 RPM to an $OD_{600}$ of ~ 1.5 – 2.0 as described (2.2.1). Cells were then harvested and successively concentrated over three runs to 50 mL. Runs involved a spin down at 5000 RPM for 5 min followed by a pellet re-suspension in TBS. Prior to sand inoculation, 50 mL of a two-times (2X) concentrated cementation (CM) medium (2X CM; 0.5 M $CaCl_2$ [Anhydrous granular, Sigma-Aldrich], 0.5 M urea [BioReagent, Sigma-Aldrich], 5 g/L yeast extract [YE] [BD Difco™], 50 mM Tris-Base [Trizma©, Sigma-Aldrich], pH 8) was added to the final suspension. Negative controls were 1:1
mixes of $ddH_2O$ and 2X CM as well as the non-ureolytic strain *B. subtilis* (BGSC 3A1[T]) (Cruz-Ramos et al., 1997). A positive control with *S. pasteurii* (ATCC 11859), a ureolytic organism capable of ureolytic MICP, (van Paassen et al., 2009) was also run. The procedure was repeated every 24 h to provide fresh cells for injection during cementation trials.

### 2.3.3 Sample preparation and cementation trial

Triplicate test units were constructed from aluminum (Fig. 1), each housing a triplicate set of sample moulds measuring 60 x 60 x15 mm. Moulds were sized according to the sample intake for the direct shear apparatus (Model: ELE-26-2112/02) utilized in confined shear tests. Each mould was equipped with a drainage valve for

medium replacement.  Filter paper was placed over the drainage valve holes during silica sand packing to prevent material

loss. Silica (autoclaved; dry cycle, 120 $^{\circ}$C, 15 min) was packed to a dry density of 2.50 - 2.55 g/cm$^3$ and washed three times with 25 mL of TBS.  Thereafter, 25 mL of a CM suspension containing bacteria was added and incubated for 24 h.  At the end of the incubation period, the CM suspension was drained and the sand washed three times with 25 mL of TBS. This was repeated twice for a total of three, 24 h incubation periods. In addition, during each 24 h incubation period, 1 mL of solution was reserved for serial dilution at two times: (1) immediately after addition of CM suspension and (2) immediately before

draining of CM suspension. Serial dilutions were performed using TBS onto agar plates as described (2.1) with 0.1 mg/L Ampicillin (Sigma-Aldrich) to measure biomass as colony forming units (CFU).  Many species of *Bacillus* were found to be resistant at these Ampicillin concentrations (Environment Canada 2015), but otherwise lethal to most contaminant bacteria. In-lab tests observed more than 95 % survival rates for all considered *Bacillus* and *Sporsosarcina* strains compared to a less than 0.1 % survival rate among a model *E. coli* (DH5$\alpha^{TM}$ , Thermofisher). Ambient temperatures of treated sands were

maintained at 22 $^{\circ}$C, reflective of average sub-surface soil temperatures of central North American
climate in the summer (Mesinger et al., 2006).

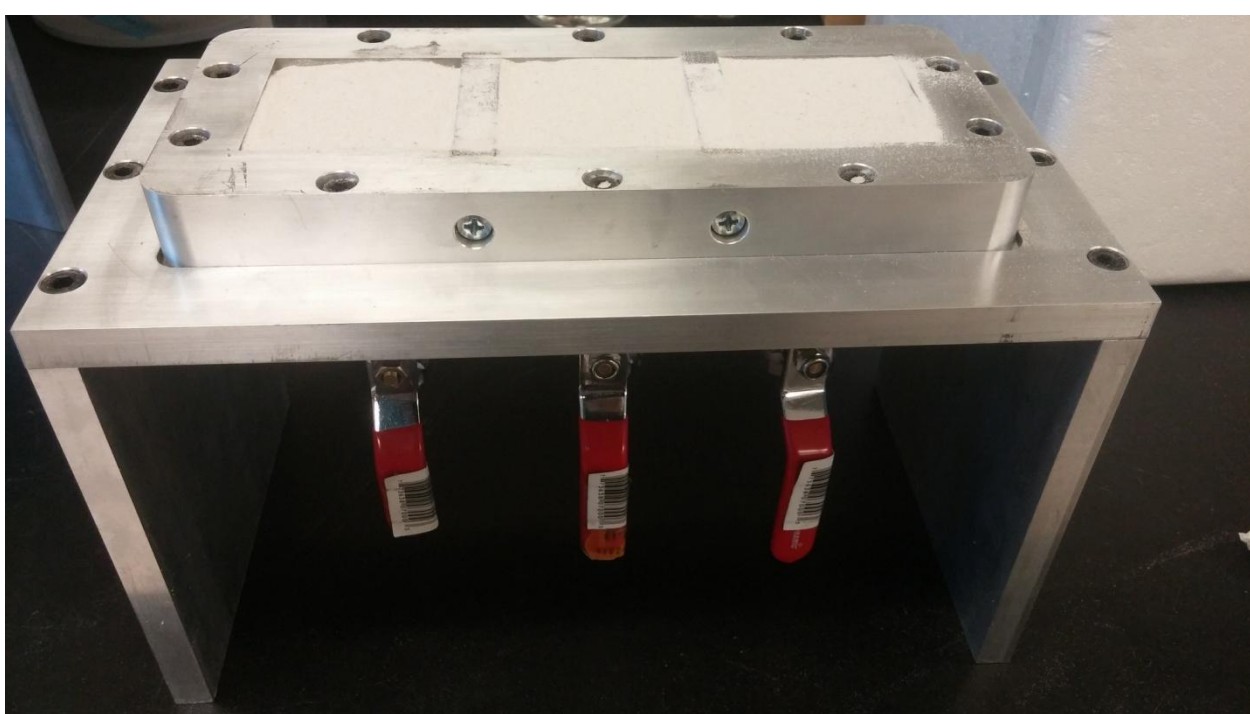

Fig. 1. Aluminum mould constructed for cementation testing


**2.3.4 Confined direct shear tests**

Treated, drained samples were washed twice with 25 mL of ddH20 and dried in an oven for 48 h at 65 ºC.  Washing with ddH2O was done to remove salts other than CaCO$_3$ to prevent cementation of the sand due to salt precipitation in the drying

process as has been found in the literature (Jia and Jian 2015 ; Zeng et al. 2018).  The shear strength tests were performed in

a direct shear machine (Model: ELE-26-2112/02). Unless otherwise specified, shear tests were performed on samples with an applied normal stress of 25 kPa. Shear stress was then applied to failure at a rate of 2.5 mm/min under dry and drained conditions. Stress-strain curves were acquired via LabView data acquisition software.

**2.4 Scanning electron microscopy (SEM) observation**

Silica grains from the surface layer of treated, washed and dried sands were mounted on a samples holder (51 mm) using double-sided copper tape and observed to confirm the crystalline nature of the resulting precipitates using a JEOL6610LV scanning electron microscope (5 kV). Elemental composition of surface structures was analysed, in parallel, by energy dispersive x-ray spectroscopy (EDS).

**2.5 Environmental simulation tests**

**2.5.1 Water flushing**

The ability for cured samples to perform following a one month saturation period was tested. Treated silica sands were incubated with $ddH_2O$ over 6 periods of incubation. Each period involved injection of 25 mL of $ddH_2O$ followed by a 5 day treatment under ambient temperature of 22 $^o$C. Volumes were replaced at the end of each period. No aliquots for colony counts were taken.

**2.5.2 Ice-water cycling**

To understand the degree to which ice cycling impacted the shear strength of treated silica sand, a selected number of samples were treated over 6 periods of $ddH_2O$ incubation as described immediately above. However, each period began with a freezing at -20 $^o$C for 24 h, holding for 3 days at -20 $^o$C, followed by a thawing for 24 h at 22 $^o$C. The selected maximum and minimum temperatures reflect those capable of being reached in Ontario winters and summer (Canada), respectively, according to Environment Canada (Climatic station: Ottawa CDA) (Government of Canada 2017).

**2.5.3 Acid erosion**

Formulation of an acid rain model was made according to average pH values (pH ~ 4.4) of rainfalls reported for North-Eastern regions of North America (Environment Canada, 2013). The final pH was adjusted using concentrated sulfuric acid ($H_2SO_4$). One delivery volume of acid rain was equivalent to the average monthly precipitation of a North American region (April, Ottawa, Canada), calculated from records of Environment Canada (Climatic station: Ottawa CDA) (Government of Canada 2017). Rain was delivered as described for 'Water Flushing' with $ddH_2O$ but for a single incubation period. Following incubation, the treated volumes were flushed with 25 mL of $ddH_2O$.

**2.6 Statistical processing**


All statistical manipulations were performed in Excel (2007). Sample means were reported alongside the standard error of the mean (SE) or standard deviation (SD). Normality of all data sets were confirmed with the Anderson-Darling test ($\alpha = 0.05$). The Student's t-test (unpaired, two-tailed; $\alpha = 0.05$) was utilized to compare sample means of experimental conditions for statistical significance. Prior to each t-test, homogeneity of variances for data sets were

determined using a F-test ($\alpha = 0.05$). Where variances were statistically observed as unequal, a Welch's t-test was adapted to test statistical significance between two sample means.

**3 Results**

**3.1 $NH_3$-$NH_4^+$ production**

Among the different bacterial strains considered, *S. pasteurii and S. ureae* were capable of producing the first and second highest levels of $NH_3$-$NH_4^+$, respectively, per unit of time, in both UB-1 (32.50 U/mL ; 29.00 U/mL) and UB-2 (32.76 U/mL ; 30.28 U/mL medium (Fig. 2a, 2b). Isolates of *B. subtilis* (2.91 U/mL), *B. megaterium* (4.87 U/mL) and *L. sphaericus* (5.89

U/mL) displayed a lower peak of $NH_3$-$NH_4^+$ production in both media. When urea in medium moved from the sole source (i.e., UB-2) to one of a number of sources (i.e., UB-1) for nitrogen, $NH_3$-$NH_4^+$ production dropped to near zero values (Fig. 2a, 2b) for *B. subtilis* (0.44 U/mL), *B. megaterium* (0.56 U/mL) and *L. sphaericus* (1.20 U/mL) that were statistically significantly different ($p < 0.05$, $n = 6$) from the final UB-1 values for each species. However, isolates of *S. ureae* and *S. pasteurii* observed no statistically significant difference ($p > 0.05$, $n = 6$) between final values recorded in UB-1 and UB-2

medium. Instead, a rise in production ($t = 0 - 5$ h) followed by a levelling off in value ($t = 6 - 12$ h) was the general trend observed in UB-1 and UB-2 medium (Fig. 2a, 2b).



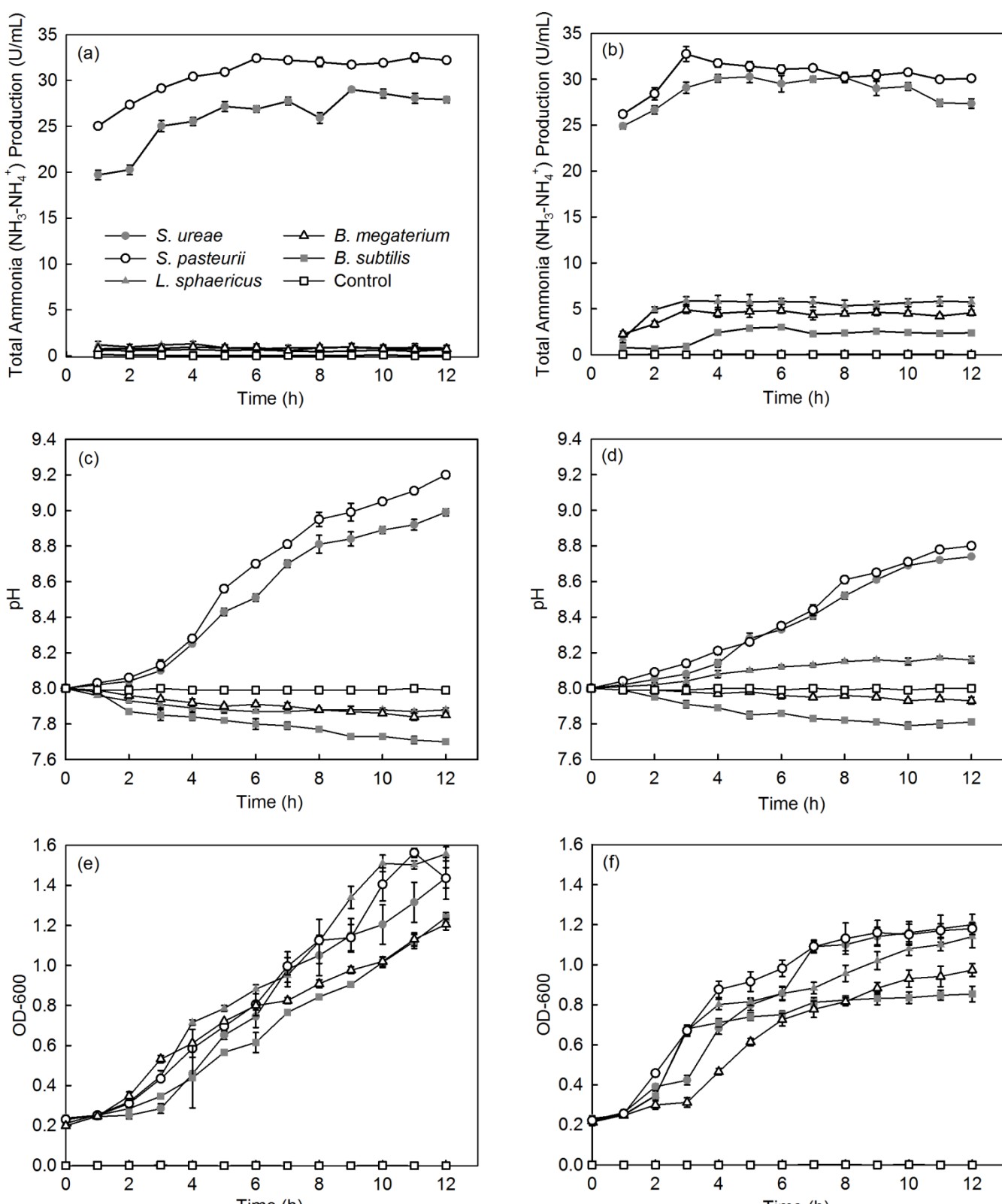

Fig. 2. **(a)**, **(b)** NH$_3$-NH$_4^+$ production (U/mL = umol of NH$_3$-NH$_4^+$ / minute.mL.OD$_{600}$ of culture) ; **(c)**, **(d)** pH ; and **(e)**, **(f)** growth of selected bacteria types in **(a)**, **(c)**, **(e)** UB-1 (*No yeast extract [YE]*) and **(b)**, **(d)**, **(f)** UB-2 (*10 g/L YE*) nutrient conditions (*SD, n = 6*). YE was a nitrogen source in the growth medium.

### 3.2 Examination of colony abundance in culture

All strains showed a decline in growth progression when medium was restricted (i.e., UB-2) to urea as nitrogen and glucose as carbon, sources, respectively (Fig. 2e, 2f). Growth repression was greatest in the cases of *B. subtilis* (-33.9 %), *L. sphaericus* (-26.8 %) and *B. megaterium* (-23.6 %) compared to *S. pasteurii* (-17.8 %) and *S. ureae* (-16.6 %). Additionally, the final $OD_{600}$ (t = 12 h) achieved for all strains in UB-2 medium was decreased compared to UB-1 medium values (t = 12 h) and the difference in value for each strain was found to be statistically significantly different ($p < 0.05$, n = 6). Growth cessation (i.e. stationary phase) occurred for *S. ureae* and *S. pasteurii* in both conditions but later in UB-1 (t = 11 h) compared to UB-2 (t = 9 – 10 h) medium (Fig. 2e, 2f); they grew logistically in both medium conditions. In general, growth of *L. sphaericus*, *B. subtilis* and *B. megaterium* in UB-2 medium followed a logistic growth curve too. However, in UB-1 medium their growth fit an exponential model, whereby an exponential growth phase was observed from t = 4 – 12 h following a lag phase of growth between t = 0 – 3 h.

### 3.3 Changes in pH

The alkalinity increased with the increase in time for the strains of *S. ureae* and *S. pasteurii* studied, in both UB-1 (8.99, 9.2) and UB-2 (8.74, 8.8) medium. The lowest final pH values were observed in *L. sphaericus* (7.88; 8.16), *B. megaterium* (7.85 ; 7.93) and *B. subtilis* (7.70 ; 7.81) in UB-1 and UB-2 medium, at the end of 12 h (Fig. 2c, 2d). While pH continued to rise for *S. pasteurii* and *S. ureae* in either UB-1 or UB-2 medium, it was constant for *L. sphaericus*, *B. megaterium* and *B. subtilis* after time in UB-1 medium as early as 6 h (*L. sphaericus* and *B. megaterium*) in UB-2 medium. While final pH values for *L. sphaericus*, *B. megaterium* and *B. subtilis* reached higher final (t = 12 h) values in UB-2 medium compared to UB-1, that were found to be statistically significantly different ($p < 0.05$, n = 6), the opposite was true for *S. pasteurii* and *S. ureae*; values in UB-2 were lower compared to UB-1 and the difference was found to be statistically significantly different for each species ($p < 0.05$, n = 6). In general, acidity increased with the increase in time for *L. sphaericus*, *B. megaterium* and *B. subtilis* in UB-1 medium. This was also true in UB-2 medium except for *L. sphaericus* which showed an increase in pH over time.

### 3.4 Mechanical and biological behaviour in MICP reinforced sands

Experiments of sand consolidation with triplicate holding vessels (Fig. 1) mixed with *S. ureae* (135.77 kPa) or *S. pasteurii* (135.5kPa) and fed MICP medium (i.e., CM-1) had improvements in their direct shear strength compared to control vessels (15.77 kPa) fed with MICP medium only. In fact, the difference in direct shear strength values for *S. ureae* and *S. pasteurii* compared to control vessels were found to be statistically significantly different ($p < 0.05$, n = 3). However, the difference in strength between *S. ureae* and *S. pasteurii* were not statistically significantly different ($p > 0.05$, n = 3). Mixtures of non-ureolytic *B. subtilis* (28.1 kPa) showed no statistically significant difference ($p > 0.05$, n = 3) in value when compared to the control (Fig. 3). While pre-injection (21.9 x $10^7$ CFU/mL) and post incubation (3.2 x $10^7$ CFU/mL) cell abundance was highest in the case of *B. subtilis*, (Fig. 4) all bacterial isolates showed a decrease in cell abundance when comparing pre-injection to post incubation cell abundance with statistically significant differences ($p < 0.05$, n = 9). Also, the percentage loss of cell abundance, taken as the difference between post incubation and pre-

375     injection cell abundances divided by the initial pre-injection cell abundance (-77.7 % [*S. ureae*], -75.4 % [*S. pasteurii*], -77.7 % [*B. subtilis*]) were not statistically significantly different (p > 0.05, n = 9) when comparing values between species.  Of note, the medium-only control had no cell growth (CFU/mL) observed before and after incubation.

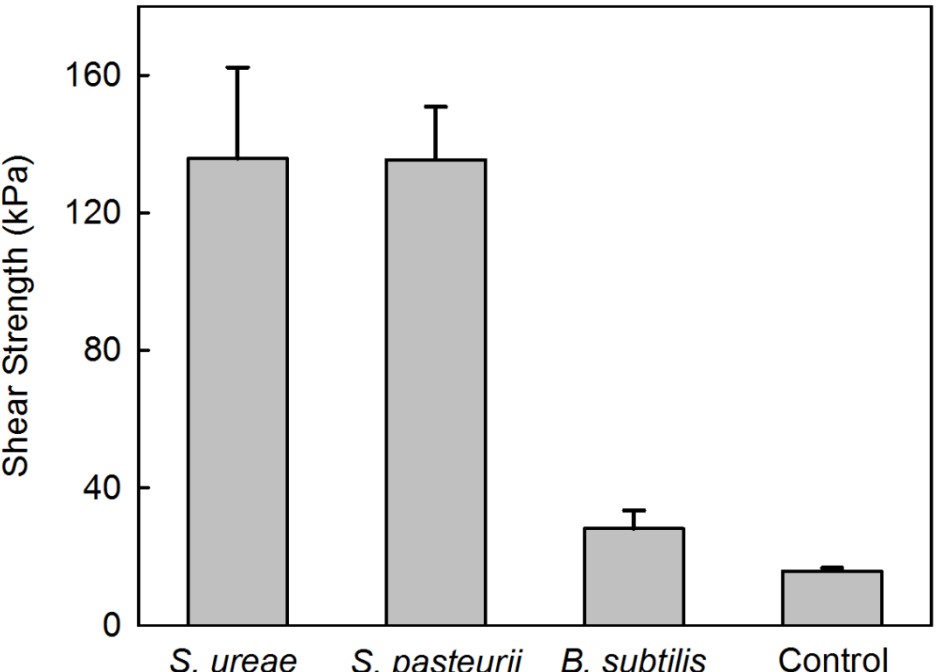

Fig. 3. Direct shear strengths ($\tau$, *kPa*) of treated sands (*SE, n = 3*).

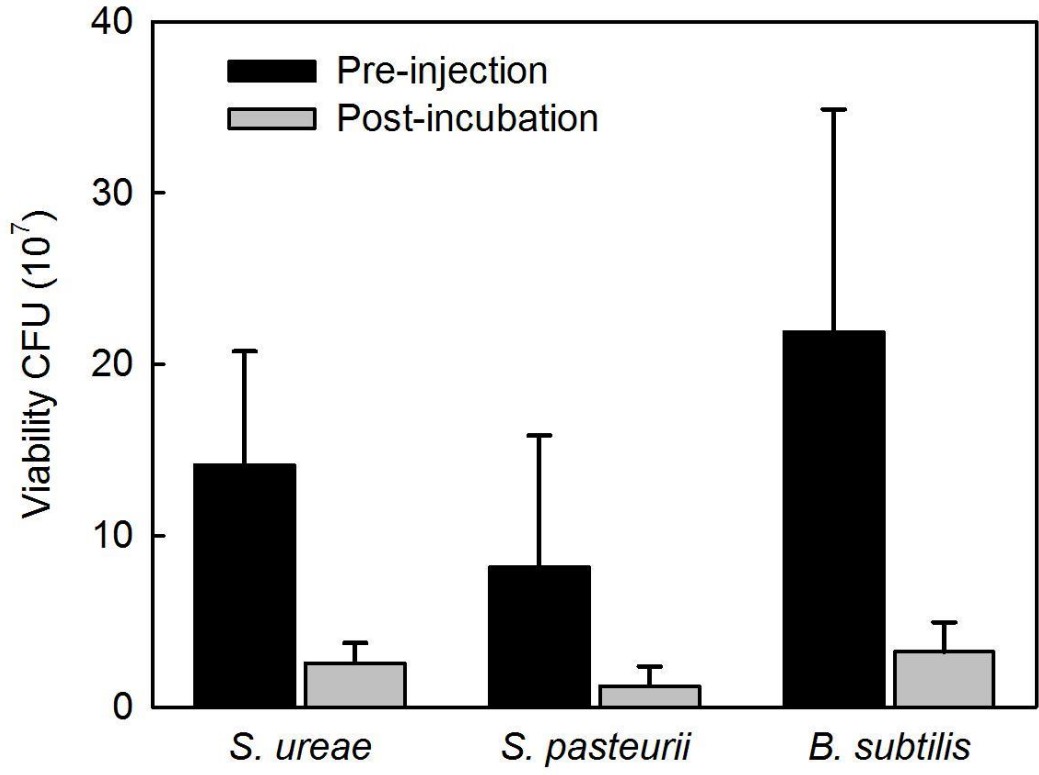

Fig. 4. Microbial viability of treated sands before injection (*black bars*) and after incubation (*gray bars*) (*SD, n = 9*).

### 3.5 Microstructure investigation

The precipitation of calcium as $CaCO_3$ via MICP was visualized. Sand granules from approximately the first 1cm of sands treated with MICP solution (i.e., CM-1) combined with *S. ureae* are shown (Fig. 5a, 5b) where crystals arranged in rosette peaks (20 – 40 μm) can be seen across the surface of a sand grain (Fig. 5a, 5b). Rod-shaped structures (40 – 80 μm) can also be visualized, though less commonly, across grain surfaces (Fig. 5a, 5b). Calcium, carbon and oxygen peaks captured by EDS analysis for crystals organized in 'rosette' patterns as well as in rod-shaped structures suggest $CaCO_3$

precipitation (Fig. 5c, 5d).

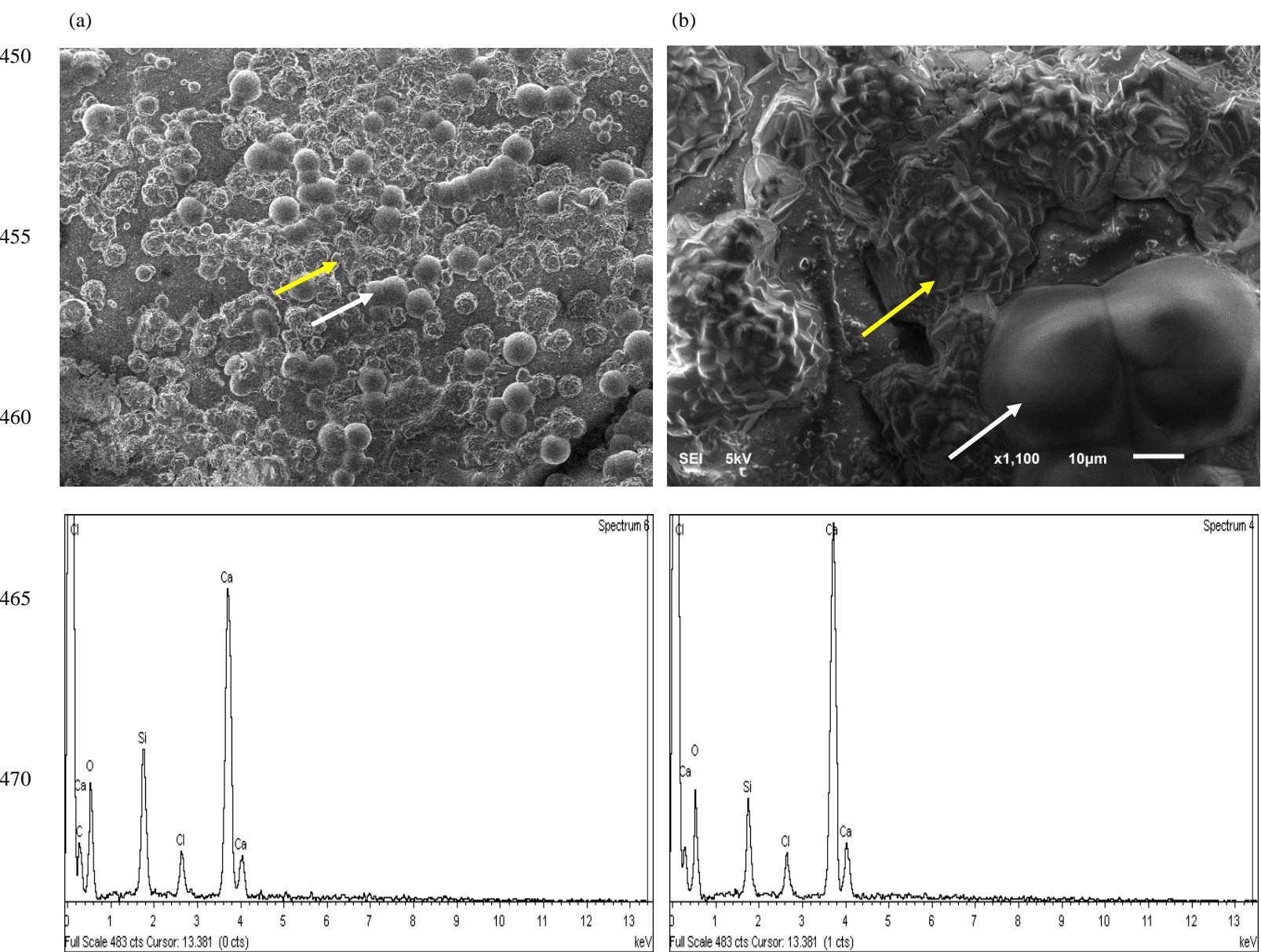

Fig 5. SEM image of the (a) whole surface (*bar, 100 μm*) and (b) magnified (*bar, 10 μm*) silica granule with crystalline (*yellow arrow*) and amorphous (*white arrow*) calcium structures following bacterial treatment. EDS analysis shows the chemical composition of (c) crystalline and (d) amorphous precipitates.

**3.6 Environmental durability of MICP**

A reduction in the reinforcement of sands by $CaCO_3$ mineralisation with *S. ureae* inoculations was observed following exposure to acid rain as direct shear strengths reduced to 39.7 kPa (Fig. 6) or 29.2 % compared to those with no such treatment (Fig. 3).  Treated sands under flooding (111.7 kPa) or freeze thaw (93.5 kPa) rounds had better durability (i.e., strength retention) compared to acidified states, with differences in strength being statistically significantly different ($p <$ 0.05, n = 3). In fact, no severe mechanical damage was incurred by samples treated under simulated flooding or freeze-thaw cycles (Fig. 6); when comparing the difference in their direct shear strengths to sands tested under ideal (i.e., non-environmental) conditions, these differences were found to be not statistically significantly different ($p > 0.05$, n =3) (Fig. 3).

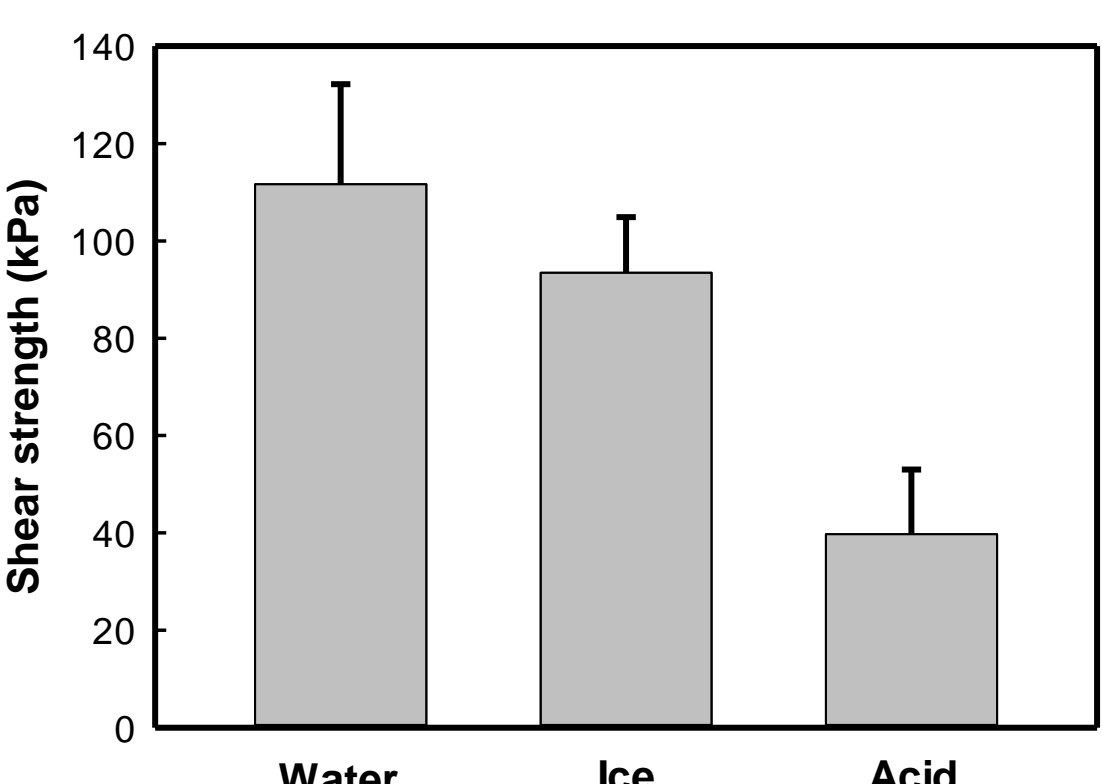

Fig 6. Direct shear strengths (*τ, kPa*) of treated sands with *Sporosarcina ureae* in flood (*water*), freeze-thaw (*ice*) and acid rain (*acid*) simulations (*SE, n = 3*).

## 4 Discussion

In characterizing *S. ureae* as a ureolytic organism in MICP, the goals of the study were to understand: (1) its ability to degrade urea over time relative to other commonly applied MICP bacterial isolates and (2) its preference for urea as a nitrogen source. The strain (BGSC 70A1) was consistent in its total nitrogen ($NH_3$-$NH_4^+$) production regardless of whether the nutrient medium included (i.e. UB-1) or did not include (i.e. UB-2) yeast extract. This can be attributed to mostly urea catabolism in UB-1 medium and entirely so in UB-2 medium as urea was the sole source of nitrogen. It is important to note that minor mineralization of the yeast extract components in UB-1 medium would likely have contributed ammonium (Gat et al., 2014) in this medium condition. This is supported by data recorded for the negative control (medium-only) in UB-1 medium with production as high as 0.12 U/mL (Fig. 2a, 2b). Also, degradation of amino acids from bacterial metabolism, such as ornithine, particularly supplied in UB-1 medium via yeast extract, could also contribute to total nitrogen in solution for this condition (Cruz-Ramos et al., 1997). For both media (UB-1 and UB-2) dissolution of ammonium as ammonia into the atmosphere would have reduced available nitrogen for measurement, over time. Thus, a quantitative urea hydroylsis rate cannot be determined from the data collected, as nitrogen production over extended periods of time is a complex collection of some or all of these processes. This limits the conclusions able to be drawn as only the broad bacterial activity in medium, as regards preferences for urea as a nitrogen source, overtime can be considered. For a quantitative method determining urease rates a robust protocol is presented by Lauchnor et al. (2015). Also, urea hydrolysis-induced $CaCO_3$ precipitation rates can be determined by measuring the decrease in dissolved $Ca^{2+}$ ions overtime (Harbottle et al., 2016). However, overall, the total nitrogen production over time draws support for *S. ureae* as a promising MICP candidate in biocement as over the time period measured it was able to produce a consistent amount of nitrogen as ammonia- ammonium in UB-1 or UB-2 medium and ammonia production has been found to be directly proportional to $CaCO_3$ production (Reddy et al., 2010) and soil stabilization (Park et al., 2012). As mentioned, the production of nitrogen by *S. ureae* in medium is due mostly, or completely, to urea catabolism and this process is likely driven chiefly by its urease enzyme (Gruninger and Goldman, 1988 ; Mobley and Hausinger, 1989). Alternatively, an unknown urea-degrading enzyme other than urease could produce or contribute to the result. Notably, all *Bacillus* strains observed a decrease in total ammonia production when yeast extract was available (i.e., UB-1). This was not observed for *S. ureae* much like *S. pasteurii*. Urea is a nitrogen source for bacterial growth, often catabolised by urease, (Lin et al., 2012) which has been found to be controlled by nitrogen levels and pH as well as other factors which can differ between bacterial species (Mobley et al., 1995; Mobley et al., 2001). Our observations indicate that *S. ureae* selects for urea in a metabolic pattern potentially similar to *S. pasteurii* and quite differently from the *Bacillus* strains investigated here, which appear to have medium-dependent metabolism of urea.

The observation that the investigated *Bacillus* strains have medium-dependent metabolism of urea is particularly interesting for *B. subtilis* as it has been applied as a non-ureolytic control organism in previous literature (Stocks-Fischer et al., 1999; Gat et al., 2014). In UB-2 medium, a non-zero total ammonia activity was measured for this strain (Fig 2a, 2b). This is consistent with previously published literature linking total ammonia production to urea breakdown from urease, when urea is the sole source of nitrogen and urease is the assumed main catabolic enzyme; the enzyme expressed constitutively in species of *Sporosarcina* (Mobley et al., 1995) but in a repressible manner (i.e., activated in the absence of $NH_4^+$ and other

forms of nitrogen [i.e., $NO_3^-$] and urea being the sole nitrogen source) in strains such as *B. megaterium* (Mobley and Hausinger, 1989) and *B. subtilis* (Atkinson and Fisher, 1991; Cruz-Ramos et al.,1997). This is indeed suggested by our data as it was observed for *B. subtilis* (and also for *B. megaterium* and *L. sphaericus*) that increased total ammonia production

reached higher values in UB-2 medium compared to near zero values in UB-1 medium with yeast extract as an alternative nitrogen source. In fact, in UB-2 medium peaks were reached within 3 – 6 h from near zero values (t = 0 – 1 h) for all *Bacillus* species, further suggesting an increase in processes related to urea hydrolysis, such as urease expression, overtime following a reduction in genetic repression (Fig 2a, 2b). This also corroborates well with growth patterns. A comparatively slow growth rate occurred (t = 8 – 12 h) after a comparatively fast (t = 3 – 7 h) rate of growth following a lag period (t = 0 –

2 h) for these strains, in general (Fig. 2a, 2b). An increase in urease, or other urea hydrolysis processes, may account for an ability to grow quickly (t = 3 – 7 h) despite nitrogen limitation in UB-2, as ureolysis would provide nitrogen for growth related processes. However, growth could have been restricted, overtime, due to other nutrient limitations such as glucose depletion. This would explain a continued but reduced growth rate (t = 8-12 h) (Fig. 2a, 2b). Alternatively, or in addition, the decreased growth could be due to decreased dissolved oxygen content in medium overtime, which is required for aerobic

respiration, such that each *Bacillus* species switched to a slower, anaerobic growth pattern. An increase in harmful metabolites such as organic acids in solution over time could also have hindered growth; they are supported to have occurred for these species in UB-1 medium as a decrease in pH over time was observed which correlates to organic acid production (Fig. 2c). Taken together, this has significance as while *B. megaterium* and *L. sphaericus* have been investigated as candidates in ureolytic MICP, this has not been extensively the case for *B. subtilis* which in this study shows ureolytic

capability under specific conditions. This may guide future research on ureolytic MICP with *B. subtilis*, particularly where cementation media do not contain nutrient rich additives such as yeast extract. This has been the case in some literature solutions for inducing ureolytic MICP (van Paassen et al., 2010; Cheng et al., 2013). In this study *B. subtilis* was included in sand solidification as a non-ureolytic strain control as the cementation medium contained yeast extract, intended for maximum biomass support and $CaCO_3$ production rates (van Paassen et al., 2010).

It is clear that *S. ureae* prefers an alkaline environment, like *S. pasteurii* and quite different from the other isolates in trials, as in both growth conditions samples grew not only exponentially but towards an increased pH (Fig. 2c, 2d). Urea hydrolysis, driven potentially by urease, in this species, may maintain ureolytic activity for production of the highly alkaline environment to which it is suited for growth as an alkaliphile and for its role as a nitrogen cycler (Gruninger and Goldman 1988). These conditions are also important for $CaCO_3$ production (Whiffin et al., 2007). It can also use the charge gradient

generated from ammonium production for energy (Jahns 1996) to support growth. A diagram of this ATP-generating system coupled to ureolysis is available in the work of Jahns (1996) and Whiffin (2004). Additionally, the ammonium is an accessible nutrient (i.e. nitrogen) source (Gruninger and Goldman 1988). This may partly account for *S. ureae* and *S. pasteurii* having the smallest change in growth between UB-1 and UB-2 medium by having the material but also energetic means to multiply. This is extremely promising as van Paassen et al. (2010) determined the $CaCO_3$ precipitation rate is

positively correlated to the number of viable micro-organisms in solution. Thus, taken together, the ureolytic, pH and growth data of this study support *S. ureae* as superior in ureolytic action to every *Bacillus* strain considered except *S. pasteurii*. Indeed, the work of Harbottle et al. (2016) likewise found *S. ureae* and *S. pasteurii* to be about as efficient in terms of ureolytic activity (2016). Given the current data, *S. ureae* and *S. pasteurii* are comparable as candidates for ureolytic MICP.

This should prompt interest for further investigations differentiating between the two strains on such parameters as protease

activity, exopolysaccharide production and biofilm levels, also connected to MICP capability (Achal et al. 2010), so as to

identify the superior candidate. Some differential work has already been done (Sarmast et al., 2014).

To understand the macroscopic engineering aspects of *S. ureae* in MICP application, efforts of this study were focused on

measuring and assessing its ability to strengthen model sands via urea hydrolysis to form $CaCO_3$. In experiments with a

model silica sand featuring poor geotechnical characteristics (i.e., uniform sand profile) for high susceptibility to settling and

static strength decreases (Conforth 2005), it was clearly shown that the *S. ureae* treatment led to consolidation of the medium

in 48 h with an improvement in strength to 135.77 kPa.  This was eight times that of the control treatment (15.76 kPa) (Fig.

3).  In addition, while average consolidation strengths had no statistically significant difference ($p < 0.05$, n = 3) between *S.*

*ureae* and *S. pasteurii*, the peak sample strength recorded for a *S. ureae* mould (175.8 kPa) exceeded the maximum sample

strength recorded for *S. pasteurii* (165.7 kPa), the typical model ureolytic organism in MICP soil strengthening.  It was also

well above peak average strength recorded for *B. subtilis* (28.1 kPa) (Fig. 3). This is as expected; *B. subtilis* is a non-

ureolytic organism in the 'good nitrogen' (Atkinson and Fisher, 1991) nutrient conditions supplied by the yeast extract of

CM-1 medium. Other *Bacillus* species were not tested under the assumption that they too would experience repressive urea

hydrolysis expression in CM-1 medium and would produce similar observations as a result. This is supported by data

provided by the groups of Al-Qabany et al. (2012) and van Paassen et al. (2010) that found $CaCO_3$ precipitation, and by

inference soil strength, improved with more suitable micro-organisms in MICP.  Taken together, this study provides

evidence that *S. ureae* is capable of soil improvement by ureolytic MICP similarly to *S. pasteurii*.

The presence of crystals as organized 'rosettes' and amorphous 'rods' was observed (Fig. 5a. 5b) along sand granules treated

with *S. ureae* and are evidence that it is capable of inducing prevalent formation of secondary minerals. The structures were

analyzed by EDS and the results provide support for $CaCO_3$ formation (Fig. 5c, 5d). Assuming that the solution was

saturated with respect to $CaCO_3$ and that the nucleation and crystallization of the calcite polymorph was thermodynamically

favoured overtime, the organized deposits should represent calcite (De Yoreo and Vekilov, 2003). However, fast nucleation

and crystallization can result in amorphous $CaCO_3$ structures and could explain the 'rod' deposits that appear amorphous in

morphology under SEM (Fig. 5a, 5b) (Addadi et al., 2003). This observation is limited though as SEM cannot discriminate

among $CaCO_3$ polymorphs which can have varying morphology based on the crystallization conditions (Ni and Ratner,

2008). The exact polymorph of $CaCO_3$ for each structure could be distinguished with techniques such as x-ray diffraction

(XRD) and / or Fourier transform infrared spectroscopy (FTIR) (Anthony et al., 2003; Ni and Ratner, 2008). Assuming the

'rod' structures are amorphous precipitates, this indicates that the treatment conditions were potentially sub-optimal for the

maximum precipitation of crystalline $CaCO_3$ such as calcite overtime. This could be due to high, local chemical

concentrations (e.g., calcium) which have been found to hinder $CaCO_3$ crystal formation as calcite (Al Qabany et al., 2012).

Investigators may be prompted to test alternative calcium concentrations from those used in this study for injections so as to

increase the efficiency of crystalline $CaCO_3$ precipitation in MICP.  Finally, medium and *B. subtilis* treated sands gave no

discernible crystal $CaCO_3$ formation (data not shown). This provides evidence of superficial strengthening in shear tests for

these treatments based on natural biofilm excretion (*B. subtilis*) or sporadic mineral crystallization.  Thus, overall, the microscopy

evidence does support that *S. ureae* can precipitate $CaCO_3$ for strength improvements in soil which was part of the goal in
studying *S. ureae* in MICP.

Analysing the cell viability of injections before and after incubation in treated sands, it was found that *S. ureae* maintained

higher post-incubation ($2.56 \times 10^7$ CFU) cell abundance compared to *S. pasteurii* ($1.21 \times 10^7$ CFU) and that these differences

were statistically significant ($p < 0.05$, $n = 9$) (Fig. 5). Also, both species' cell abundance was lower and found to be

statistically significantly different ($p < 0.05$, $n = 9$) compared to the cell abundance for *B. subtilis* ($3.2 \times 10^7$ CFU). This

difference could be due to the solution (i.e., TBS) utilized for serial dilution of the growth medium.  The TBS did not include

ammonium and was not buffered at a high pH which are two necessary conditions for the survival of alkaliphilic species

such as *Sporosarcina* (Morsdörf and Kaltwasser, 1989). Thus, a deflated value for *S. pasteurii* and *S. ureae* would result.

Also, moulds become mostly anaerobic overtime below the subsurface and within the microenvironments of sand grains as

oxygen is depleted by bacterial respiration (van Paassen et al., 2010). *B. subtilis* cells may have survived anaerobically

(Clements et al., 2002) as opposed to the obligate aerobes *S. ureae* and *S. pasteurii* (Claus and Fahmy, 1986), leading to

higher post incubation cell abundance for *B. subtilis*.  However, considering the percentage loss of cell abundance calculated

as described (3.4) is comparable between all three species this indicates that neither species outperforms the other in cell

survival while in the high salt, high urea CM-1 medium with incubation in treated sands.  That being written, the total cell

abundance in *S. ureae* is higher compared to *S. pasteurii*.  This is important as cells provide nucleation points for $CaCO_3$

formation. Indeed, the literature reports designate that strength enhancement by ureolytic MICP is driven by urea hydrolysis

activity but also by the presence of bacteria acting as nucleation sites (Stocks-Fischer et al., 1999; Gat et al., 2014).  While

sand surfaces can also act as nucleation points, the negatively charged bacteria cell wall attracts positively charged cations

(e.g., calcium) preferentially for the controlled nucleation of $CaCO_3$ over time. In fact, it has been shown that cell abundance

in MICP treatments positively correlate to the precipitation of $CaCO_3$ in both the rate of production and crystal size (Morris

and Ferris, 2006). The group of Hommel et al. (2015) have even developed a model showing that calcite precipitation is

proportional to cell abundance (i.e. biomass) and potentially improved soil strengths. This model assumes that the features of

the cells such as biofilm production around their cell walls favour and facilitate the precipitation of $CaCO_3$. It follows that

any intact cell wall part of the biofilm can facilitate the precipitation process whether the cell itself is alive or dead. Thus, in

general, more cells equates to more $CaCO_3$ precipitation. However, in this study, *S. ureae* gave rise to strengths in sands that

were not statistically significantly different ($p > 0.05$, $n = 3$) versus *S. pasteurii* treatments.  This is unexpected since *S. ureae*

had comparable ureolytic activity to *S. pasteurii* but higher cell abundance overtime in precipitation medium. Therefore,

more $CaCO_3$ precipitation should have occurred and led to a greater strength increase in sands in *S. ureae* treatments. This

non-linear increase in strength compared to cell abundance can be a result of a number of factors. For example, the ability for

cells to precipitate $CaCO_3$ can be hindered when an abundance of cells injected into porous material (i.e., sands) lead to pore

plugging from the organic matter (i.e., cells). This has been seen to lead to a varied amount of $CaCO_3$ precipitation

throughout the volume of a mould (van Paassen et al., 2009).  Where cells are distributed more evenly they can facilitate the

precipitation of $CaCO_3$ as nucleation points (Hommel et al., 2015).  This may explain why *S. ureae*, having a comparable $NH_3$-

$NH_4^+$ activity to *S. pasteurii* did not outperform it on average in undrained, direct shear strength tests despite having a higher

cell abundance on average. It may also explain the broader range of strengths achieved in *S. ureae* (Fig. 3). For example, a

sub-optimal spreading mechanism could have hindered strength achievement in some moulds of *S. ureae* treatment where pore plugging by organic matter (i.e., cells) occurred.  This in mind, optimization of treatment protocols would help to determine whether or not *S. ureae* is the superior candidate compared to *S. pasteurii* given that it has consistently increased total cell abundance (Fig. 3) to support more nucleation of $CaCO_3$ overtime, in tandem with a $NH_3$-$NH_4^+$ production comparable to that of *S. pasteurii*.  However, it is important to note that *S. ureae* cells are significantly

smaller than cells of *S. pasteurii*  (Claus and Fahmy, 1986). Therefore, the total cellular surface area available for nucleation of $CaCO_3$ would be similar for the two species. This provides a possible explanation for why no statistically significant differences in strength was observed because if total cellular surface area was most important for precipitating $CaCO_3$ this means there would be no difference in strengths expected for the same total cellular surface area whether it was spread over a relatively high number of smaller cells (i.e., *S. ureae*) or fewer number of larger cells (i.e., *S. pasteurii*).

It was the current authors' focus to also apply tests in conditions reflective of a Canadian environment with a relatively novel bacterial isolate (*S. ureae*).  Sands treated with *S. ureae* and which underwent short-term flooding (111.67 kPa) or freeze-thaw cycling (93.47 kPa) showed no statistically significant ($p > 0.05$, n = 3) strength difference compared to in-lab (135.77 kPa) conditions (Fig. 6).  It has been shown that MICP treated sands maintain some porosity in materials (Cheng and Cord-Ruwisch 2012; Chu et al., 2012) and that good strength maintenance in seasonal water saturation and freeze-thaw is possible

with porous materials (Cornforth 2005).  Further studies may wish to investigate the permeability of hardened sands via *S. ureae* at various levels of $CaCO_3$ precipitation to strike a balance between porosity, peak strength and endurance overtime in weather simulations.

Predictably, it was seen that the acid rain model, reflective of a Northern Ontario rain pH (4.4), eroded the shear strength of sands (Fig. 6) to 35.5 % of originally observed values (Fig. 3).  This is a result of the reaction of acid

with $CaCO_3$ producing units of $H_2O$, $CO_2$ and salt, known as weathering.  A study by Cheng and Cord-Ruwisch (2013) reported similar results with a *Bacillus sphaericus* model. This prompts the idea that a MICP strength model, regardless of the bacteria treatment selected (*S. ureae*, *S. pasteurii*, etc.) for strength enhancement, would require a time-based repair of treated volumes. This realistically limits its geotechnical and economical practicality in the industry. However, it does prompt interest to test the ability of natural buffers, such as limes and sodas, to increase

the life-span of MICP induced strength enhancement by reducing acid rain degradation.

**5 Conclusions**

This study has worked to verify that *S. ureae* is a suitable organism to be applied in the soil hardening technology

currently being developed via ureolytic MICP.  The authors designate it a close ureolytic MICP candidate, in performance, to the well-studied *S. pasteurii* and a superior one to several other *Bacillus* strains. As larger scale simulations are employed, it is strongly encouraged by the authors that further optimization in the treatment procedure, regardless of the MICP organism selected, be undergone including ideal soil buffering to reduce certain climatic effects (i.e., acid rain) and optimum volume porosity in the space to be treated to assure an economical

application in industry.

## 6 Compliance with Ethical Standards

Funding: The study was funded by NSERC (Discovery grant number 2016-2021 ; Discovery grant number 2015-2020) and the University of Ottawa (UROP grant 2012 ; USRA grant 2014)


## 7 Competing Interests

The authors state they have no conflict of interest.

## 8 Acknowledgements

The authors would like to acknowledge the University of Ottawa (UROP Grant) and the National Sciences and Engineering Research Council of Canada (NSERC USRA and NSERC Discovery grants to D. Fortin and S. Vanapalli) for financial provisioning in support of this project. Thanks are also given to Mr. Jean Celestin, Mr.
Yunlong (Harry) Lui,  Mr. Penghai (Peter) Yin, Dr. Nimal De Silva, Dr. Erika Revesz and Mr. George Mrazek; each providing assistance in shear measurements, microscopy and/or data analysis.

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
