# Peer review of "Improving the Strength of Sandy Soils via Ureolytic $\text{CaCO}_3$ Solidification by *Sporosarcina ureae"

_Biogeosciences, 2017_

## Referee Comment (RC1) · Anonymous Referee #1 · 30 Apr 2018

This study examined the ability of Sporosarcina ureae to improve the shear strength of sandy soils via microbially induced carbonate precipitation (MICP) . The ureolytic activity of Sporosarcina ureae was examined and compared to the activity of previously investigated species of Sporosarcina pasteurii and Bacillus. MICP by Sporosarcina ureae was shown to reinforce cemented sand and increase the shear strength the material, at levels comparable to S. pasteurii and superior to Bacillus subtilis.

I believe this paper will be interest to the engineers working on biocementation, perhaps more so than to biogeoscientists. Clearly, the hardened sands have increased strength and performed well under environmental conditions (e.g. water flushing and

ice-water cycling simulations). The observation that MICP is susceptible to acid rain is also insightful. I agree with the authors that soil buffering should be considered when applying this MICP technology in certain climate regions to mitigate the effects of acid rain.

I only have one technical comment. In Figure 2, the first time points for the S. pasteurii and S. ureae experiment yielded very high NH3-NH4+ values. What was the NH3-NH4+ concentration at time zero? There appears to be significant ureolytic activity at 1 hour (Figures 2a & b), but there was very little change in the solution pH (Figures 2c & d). Based on ureolytic reactions on page 2 (line 63), I would have expected NH3-NH4+ production to be concurrent to changes in alkalinity.

―――――――――――――――――――

---

## Author Comment (AC1) · 23 May 2018

**Author's Reponse – Review #1**

**(1)    Comments from Referee**

"I only have one technical comment. In Figure 2, the first time points for the S. pasteurii and S. ureae experiment yielded very high NH3-NH4+ values. What was the NH3- NH4+ concentration at time zero? There appears to be significant ureolytic activity at 1 hour (Figures 2a & b), but there was very little change in the solution pH (Figures 2c & d). Based on ureolytic reactions on page 2 (line 63), I would have expected NH3-NH4+ production to be concurrent to changes in alkalinity."

**(2)    Author's Response**

Thank you for your insightful comments. We have taken the time to consider, arrange and incorporate the relevant data to clarify changes in (1) pH and (2) cell density (OD600) with panels (c) – (f) of Figure 2.

In addition we have also made edits to line 130 and 155 of the original PDF file to better explain the starting cell density and method to generate NH3-NH4$^+$ production values, over time, used to generate Figure 2.

We welcome any further comments regarding the technical aspects of the project.

**(3)    Author's Changes in Manuscript**

Line 130: Starting (approximate) cell density indicated for cell cultures (t = 0 h)

Line 155: Starting point (t = 1 h) for reporting NH3-NH4$^+$ production values made explicit

Line 300: Updated Figure 2 including time point data for t = 0 h for panels (c) – (f)

[revised manuscript text omitted]

**Comment [L1]:** For further clarification of the final, starting cell density when interpreting Figure 2 panels (a) – (f).

**2.2.2 Total Ammonia ($NH_3$-$NH_4^+$), pH and growth (OD-600) aliquots**

180

To evaluate different cell parameters efficiently, duplicate aliquots (2.5mL) were taken for pH tracking, $OD_{600}$ absorbance and sample storage for $NH_3$-$NH_4^+$ analysis. In brief, first, whole aliquot volume pH was taken with a SB20 symphony pH probe (VWR). Next, a 1mL volume was removed for $OD_{600}$ reading using a BioMate 3 UV-Vis Spectrophotometer (Thermoscientific). Finally, a 500 uL sample for $NH_3$-$NH_4^+$ analysis was retrieved and diluted in

185     500 uL of ddH$_2$O and stored as described by HACH Inc. (Hach Co. 2015) with the following modifications: -20 $^o$C storage, 1 drop 5 N H$_2$SO$_4$. To avoid errors in volume delivery by micropipette, measurements were taken as mass over an analytical balance, and volumes calculated assuming a density of 1 g/mL.

**2.2.3 Spectrophotometric analysis of $NH_3$-$NH_4^+$**

190

Samples were thawed and neutralized with 5 N NaOH as described by HACH Inc. (Hach Co. 2015). $NH_3$-$NH_4^+$ measurements were then performed as outlined (HACH Co. 2015) using a portable DR2700 HACH spectrophotometer after samples were brought to a measureable range (0.01 to 0.50 mg/L $NH_3$-N). All measurements for appropriate dilutions were made by mass and corrected to volume as described above. Final

195  values were reported as units (U = mol of $NH_3$-$NH_4^+$ produced per minute) per mL of culture starting from t = 1 h.

**Comment [L2]:** Data was collected to calculate NH3-NH4 production from t = 0 h and reported as a production amount, over time (i.e., U), per mL of culture, beginning from t = 1 h as can be seen on Figure 2 0 panels (c) and (d)

[revised manuscript text omitted]

---

## Referee Comment (RC2) · Anonymous Referee #2 · 25 May 2018

The manuscript describes a study of Sporosarcina ureae and this organism's ability to catalyze MICP in sand test beds. S. ureae was compared to the model MICP bacterium, S. pasteurii, in bacterial growth, ureolytic activity and shear strength of MICP treated sand. Tests were performed to investigate the effect of flooding, freeze-thaw cycles and acid rain exposure, where only acid rain reduced the shear strength significantly. The authors conclude the S. ureae can be utilized as a model MICP bacterium and is competitive with S. pasteurii in the tests performed in this study. The manuscript presents a concise study with appropriate methods and analyses to show the applicability of S. ureae in MICP. While not completely transformative, it is a worthy contribution and the results presented will be quite interesting and useful for scientists

and engineers in the field of MICP research and applications.

Below are a few small technical comments:

Abstract line 30: This sentence is not entirely clear to an unfamiliar reader, suggested changes: "However, shear strength of samples following acid-rain simulations fell to 29.2% of control MICP samples."

Line 274 and Figure 2 a,b: Suggestion regarding the confusion around U/mL units is to simply express the rates as mol/(min-mL) throughout the MS instead of designating the parameter U. If U is used, please redefine it in Fig 2 caption, as the explanation in text was easy to miss.

Figure 6: Were the same tests performed on S. pasteurii treated samples? This data would be interesting to see alongside the S. ureae treated sands.

[Figure]

---

## Author Comment (AC2) · 31 May 2018

Please see supplemental PDF attached.

===

Author's Response – Review #2

(1) Comments from Referee #2

"Abstract line 30: This sentence is not entirely clear to an unfamiliar reader, suggested changes: "However, shear strength of samples following acid-rain simulations fell to 29.2% of control MICP samples."

"Line 274 and Figure 2 a,b: Suggestion regarding the confusion around U/mL units is to simply express the rates as mol/(min-mL) throughout the MS instead of designating the parameter U. If U is used, please redefine it in Fig 2 caption, as the explanation in text was easy to miss."

"Figure 6: Were the same tests performed on S. pasteurii treated samples? This data would be interesting to see alongside the S. ureae treated sands."

(2) Author's Response

Dear referee,

We thank you for your time reviewing our manuscript. We have considered your changes and have now made the appropriate additions to the manuscript.

Figure 6: It is correct the data represents that of S. ureae only. We underwent this testing as S. ureae is a novel ureolytic MICP model for which the environmental durability testing reported has not been previously performed, according to our literature review. However, another ureolytic MICP model used to assess similar environmental durability parameters of ureolytic MICP was also highlighted for comparison purposes (see Chenge and Cord-Ruwisch 2013 DOI: 10.1139/cgj-2012-0023). Other model species (i.e. S. pasteurii) have also been tested in environmental simulations, though differently from those reported in this manuscript (see Mortensen et al., 2011 DOI: 10.1111/j.1365-2672.2011.05065.x.). Based on the results of our study, combined with those of a previous study, it was found reasonable to assess that all model species (i.e. S. ureae, S. pasteurii, etc.) in ureolytic MICP would have similar physical outcomes under the environmental simulations performed (i.e. acid-rain degradation). Thus, no further testing on other model species was done.

For your convenience, we have highlighted the sections in the manuscript in yellow which we feel best provide rationale for the above.

We do welcome any further comments regarding how we may make the manuscript

stronger and, specifically, if the above rationale can be made more clearly in the current manuscript.

Thank you again for your time and kind efforts in review.

(3) Author's Changes in Manuscript

Line 30 (Abstract): Sentence in changed to match that suggested by referee.

Line 334 (Figure 2): U/mL defined in caption as : U/mL = mol of NH3-NH4+ / minute-mL culture

Line 99-101 ; 644-645 ; 654-657 : Highlighted text in explanation of Figure 6.

Please also note the supplement to this comment:
https://www.biogeosciences-discuss.net/bg-2017-517/bg-2017-517-AC2-supplement.pdf

**Supplement:**

**Author's Response – Review #2**

(1) Comments from Referee #2

"Abstract line 30: This sentence is not entirely clear to an unfamiliar reader, suggested changes: "However, shear strength of samples following acid-rain simulations fell to 29.2% of control MICP samples.""

"Line 274 and Figure 2 a,b: Suggestion regarding the confusion around U/mL units is to simply express the rates as mol/(min-mL) throughout the MS instead of designating the parameter U. If U is used, please redefine it in Fig 2 caption, as the explanation in text was easy to miss."

"Figure 6: Were the same tests performed on S. pasteurii treated samples? This data would be interesting to see alongside the S. ureae treated sands."

(2) Author's Response

Dear referee,

We thank you for your time reviewing our manuscript. We have considered your changes and have now made the appropriate additions to the manuscript.

Figure 6: It is correct the data represents that of S. ureae only.  We underwent this testing as S. ureae is a novel ureolytic MICP model for which the environmental durability testing reported has not been previously performed, according to our literature review. However, another ureolytic MICP model used to assess similar environmental durability parameters of ureolytic MICP was also highlighted for comparison purposes (see Chenge and Cord-Ruwisch 2013 DOI: 10.1139/cgj-2012-0023).  Other model species (i.e. S. pasteurii) have also been tested in environmental simulations, though differently from those reported in this manuscript (see Mortensen et al., 2011 DOI: 10.1111/j.1365-2672.2011.05065.x.). Based on the results of our study, combined with those of a previous study, it was found reasonable to assess that all model species (i.e. S. ureae, S. pasteurii, etc.) in ureolytic MICP would have similar physical outcomes under the environmental simulations performed (i.e. acid-rain degradation). Thus, no further testing on other model species was done.

For your convenience, we have highlighted the sections in the manuscript in yellow which we feel best provide rationale for the above.

We do welcome any further comments regarding how we may make the manuscript stronger and, specifically, if the above rationale can be made more clearly in the current manuscript.

Thank you again for your time and kind efforts in review.

(3)   Author's Changes in Manuscript

Line 30 (Abstract): Sentence in changed to match that suggested by referee.

Line 334 (Figure 2): U/mL defined in caption as : U/mL = mol of $NH_3$-$NH_4^+$ / minute-mL culture

Line 99-101 ; 644-645 ; 654-657 : Highlighted text in explanation of Figure 6.

[revised manuscript text omitted]

---

## Referee Comment (RC3) · Anonymous Referee #3 · 6 Jun 2018

The manuscript by Whitaker, Vanapali and Fortin describes work comparing several Bacillus- and Sporosarcina-species regarding their ability to consolidate (cement) a 'poorly graded' sand. The work presented presents interesting data but lacks a clear direction or message as well as easily comparable data, both within the manuscript and relative to other published literature. For instance, it is unclear how the activities (enzyme units per mL) can be truly compared with each other. The number of units is a function of the amount of urease and thus (assuming a constant urea fraction in each cell) a function of the number of cells (or the biomass (weight)) the experiment was amended with. Without normalizing the data to the amount of biomass (e.g. grams of biomass) and thus reporting the activities as U/g of biomass a direct comparison

between the different cell-types is impossible.

The authors should clearly state, which strains were investigated in this work and should also explicitly discuss the fact that Sporosarcina spp. used to be classified as Bacillus spp. See e.g. Yoon, J. H., et al. (2001). "Sporosarcina aquimarina sp. nov., a bacterium isolated from seawater in Korea, and transfer of Bacillus globisporus (Larkin and Stokes 1967), Bacillus psychrophilus (Nakamura 1984) and Bacillus pasteurii (Chester 1898) to the genus Sporosarcina as Sporosarcina globispora comb. nov., Sporosarcina psychrophila comb. nov. and Sporosarcina pasteurii comb. nov., and emended description of th." Int J Syst Evol Microbiol 51(Pt 3): 1079-1086.

Also, the group of Michael Harbottle at Cardiff University has been doing work on Sporosarcina and have provided clear evidence of S. ureae being about as efficient as S. pasteurii in regards of ureolytic activity? – see e.g. Harbottle, M., Mugwar, A.J., Botusharova, S., 2016. Aspects of Implementation and LongTerm Performance of Biologically Induced Mineralisation of Carbonates in Porous Media. Goldschmidt 2016, Yokohama, Japan.

Furthermore, the mineralogical characterizations are weak, for instance it is impossible for this reviewer to agree with the assessment of rhomobohedral[ly] shaped crystals as indicated in L 414/415; also, the statement in L418 that only calcium peaks were present in the rod shaped formations cannot be followed. This reviewer clearly sees Ca, C, and O peaks in the EDS spectra in Fig 5.

The following detailed comments will support the assessment above as well as provide examples of sections and approaches, that make this manuscript hard to review and demonstrate at least some of the deficiencies that have to be remediated prior to publication.

Abstract

L21 units need to be defined

L 143 unclear whether the OD measurements were taken in systems in which $CaCO_3$ precipitation occurred – if so, the authors need to explicitly discuss the possible influence of $CaCO_3$ precipitation on OD measurements

L 147 the authors assume a SG of 1 for the fluids – this might or might not be a good assumption depending on the concentrations of urea and calcium used.

L 151 – unclear what HACH assay was used

L 163 – should this sand indeed be described as 'uniform'?

L 166 unclear how OD was measured (what was the blank, what was the pathlength – all this must be stated 189 '3 times each at 24 h periods' – unclear

L 220-224 the authors need to check whether their statements are clear and not contradictory.

L 230/231 the authors will have to discuss the effect of drying at 65 deg C

L 236 the authors should discuss to what extent a 1 month duration can be considered 'long-term' – this reviewer thinks years to decades would be considered long-term for building materials and soil stabilization

L 274 onward – as indicated above, these data are not really easily comparable since bacterial cell densities (or biomass) were not accounted for. – culture density will play an important role in the ureolytic activity

L 344-346 AND L 355-357 it is unclear what is being compared and what the p-values are indicating (or not)

L 360-364 the authors should discuss why acidity might increase and discuss & compare these parameters and treatments in more detail. L 371-373 the discussion in these lines is weak and not well organized. – This reviewer is unable to understand what is being discussed and compared

L 488 'Destruction of MICP sands'? – what does that mean?

L490 'increase'? compared to what?

L 491-493 again, this section and statistical analyses are not clear – this reviewer might be able to see that some of the specimen retained their strength once exposed to water or freezing but strength did not 'increase' – did it?

L 521/522 this author agrees that monitoring ammonium concentrations is a mediocre way to assess urea hydrolysis rate since among other factors, ammonia volatilization and ammonium uptake can affect concentrations. Hence, many groups working on urea hydrolysis-induced CaCO3 precipitation are not using direct urea measurements using either spectrophotometric assays (e.g. Jung assay) or HPLC-based analyses. – The authors must discuss in more detail how the ammonium/ammonia-based esti-amtes of urea hydrolysis rate might have affect their assessments

L 549-553 incomprehensible section

L 566 'S. urea may use the proton gradient' – unclear what 'proton gradient' – also unclear what 'may' means here

L 572-573 what are 'co-capable candidates'?

L 581 why 'only' – only compared to what?

L 585 – without showing replicates and applying proper statistical tests, any statement comparing strengths of specimen will remain highly speculative

L 594 – still don't agree with 'rhombohedral' statement

L 597/598 'Media and S. subtilis treated sands gave no discernable CaCO3 formation' – where are those data?

L 600 what was discussed in this sentence is not supported by Figure 4 (or is it?)

L 600-604 these lines make little sense to this reviewer

L 606-610 these lines make little sense to this reviewer

L 615/616 the authors should consider and discuss that dead cells can also function as nucleation points. Plus all the sand surfaces can as well. Thus, more cells might not result in a significantly increased number of nucleation points

L 621 'gave rise to non-significantly' – again, the statistical tests used by the authors are unclear to this reviewer and it is something different to 'not statistically significantly different' – this section once again does not make much sense to this reviewer

Editorial comments (not complete, just the ones that I spent the time on noting)

Urea hydrolysis equation $CO(NH_2)_2$ not $CO(NH_3)_2$

L 79 "spahericus"

L92 "alterative"

L134 "run" vs. "incubated"

L180 "fresh sample inoculate" – what is that?

L 186: 'Each mould had equipped" !?

L 188 'Silica' or 'Silica sand'?

L 228 "Visualization . . . was carried out . . .'???

L 244 'the trials' do not 'withstand'

L 386 Fig 3 'subtilis' not 'Subtilis'

L 509: 'It was chief to understand'?

L 512: 'regardless of source nitrogen availability as yeast extract or urea'? – can't follow 518 one medium, two (or more) media

L 527 and 532 why 'see also'?

L 561 'Returning to s. ureae' ?? what does that mean?

L 565 'Whiffin'

L 605 'cell viability of inoculates' – what is an inoculate?

L 616-619 – language issues – this is almost incomprehensible

L 625/626 – language issues – this is almost incomprehensible

L 635 'may prove'? or may not prove . . .

L 639 – ' would be quite proximal'? – what does that mean?

L 647 'remain' ? – maybe 'maintain'? References have random (or not so random) spaces in them that make no sense.

―――――――――――――――――――――――

---

## Author Comment (AC3) · 17 Jun 2018

Author's Response – Review #3

(1) Author's Response

Dear referee,

We thank you for the important, thorough and helpful remarks made on the manuscript. The following is our reply to the remarks and suggestions.

For ease of following our reply to each comment, we have combined the, 'Comments' and 'Manuscript changes' sections together. Each 'comment' is followed by a reply with

the specific line-by-line changes made in the revised manuscript.

Please note that the line numbers (i.e. L21, etc) refer to line numbers in the original, unrevised manuscript. Revised line numbers (i.e. RL25) refer to the line numbers in the new, revised manuscript which have been added / changed in response to the remarks made on the unrevised manuscript.

For your convenience, we have also highlighted the sections in the revised manuscript in yellow which reflect these changes. In addition, we have provided additional comments throughout the revised manuscript for added clarity. Finally, we have made editorial changes in addition to those provided for the revised manuscript.

We do welcome any further comments regarding how we may make the revised manuscript stronger and if the rationale provided below can be made more clearly.

Thank you again for your time and kind efforts in review.

(2) Comments from Referee #3 / Changes in Manuscript

Comment 1 "The manuscript by Whitaker, Vanapali and Fortin describes work comparing several Bacillus- and Sporosarcina-species regarding their ability to consolidate (cement) a 'poorly graded' sand. The work presented presents interesting data but lacks a clear direction or message as well as easily comparable data, both within the manuscript and relative to other published literature. For instance, it is unclear how the activities (enzyme units per mL) can be truly compared with each other. The number of units is a function of the amount of urease and thus (assuming a constant urea fraction in each cell) a function of the number of cells (or the biomass (weight)) the experiment was amended with. Without normalizing the data to the amount of biomass (e.g. grams of biomass) and thus reporting the activities as U/g of biomass a direct comparison between the different cell-types is impossible."

We agree with this important, necessary clarification needed in the manuscript. The manner to which data can be compared is dependent on the number of cells. We

would expect an increase in the ability to produce ammonia-ammonium, overtime, from a higher density of cells compared to a lower density of cells. The cellular density can be commonly measured in terms of cell counts, biomass (dry weight), protein content as well as optical density. In our study we used the optical density as a measure of cell density to normalize values reported for a cell type and to compare values between cell types. The optical density values used for normalization were those measured and reported in the study (Fig. 2). This technique assumes that an increase in optical density corresponds to an increase in the number of cells and that for each cell type a given optical density value corresponds to the same amount of biomass. We have addressed this issue in the revised version of the manuscript by better clarifying the units of measure from 'mL of culture' to 'mL solution normalized to cultural density' as the optical density (i.e., turbidity) value (OD600).

In the revised manuscript please see: RL154-159 and RL331

Comment 2 "The authors should clearly state, which strains were investigated in this work and should also explicitly discuss the fact that Sporosarcina spp. used to be classified as Bacillus spp. See e.g. Yoon, J. H., et al. (2001). "Sporosarcina aquimarina sp. nov., a bacterium isolated from seawater in Korea, and transfer of Bacillus globisporus (Larkin and Stokes 1967), Bacillus psychrophilus (Nakamura 1984) and Bacillus pasteurii (Chester 1898) to the genus Sporosarcina as Sporosarcina globispora comb. nov., Sporosarcina psychrophila comb. nov. and Sporosarcina pasteurii comb. nov., and emended description of th." Int J Syst Evol Microbiol 51(Pt 3): 1079-1086."

We agree with this suggestion and have made the appropriate citation regarding the classification of Sporosarcina pasteurii as previously Bacillus pasteurii. In addition, we have also clarified the previous designation of Lysinibacillus sphaericus as Bacillus sphaericus.

In the revised manuscript please see: RL108 and RL110

Comment 3 "Also, the group of Michael Harbottle at Cardiff University has been doing

work on Sporosarcina and have provided clear evidence of S. ureae being about as efficient as S. pasteurii in regards of ureolytic activity? – see e.g. Harbottle, M., Mugwar, A.J., Botusharova, S., 2016. Aspects of Implementation and LongTerm Performance of Biologically Induced Mineralisation of Carbonates in Porous Media. Goldschmidt 2016, Yokohama, Japan."

We agree with this literature suggestion and thank the reviewer for this very helpful contextual literature source. We have also included the work of, 'Sarmast et al., 2014', which while cited in the original manuscript was not appropriately incorporated into the discussion section as regards work previously done comparing S. ureae and S. pasteurii. In this study we used S. ureae based on a random selection of the 7 species of Sporosarcina distinct from Sporosarcina pasteurii and that were urease positive. It was deemed that there is a lack, though not non-existent, amount of investigation present in the literature using the species as a model ureolytic bacteria in MICP. We believe this investigation does provide a novel understanding of the use of S. ureae as a model ureolytic species in MICP as regards to comparing it to not only S. pasteurii but also other species of Bacillus. Also, it has been applied in MICP treatment in a somewhat unique sense under environmental simulation testing. Finally, we believe that the strain employed (BGSC 70A1) has a potentially improved ability in MICP relative to the strains used in previous studies as both Harbottle et al. (NCIMB9251) and Sarmast et al. (PTCCi 1642) found that the ureolytic activity of S. ureae was less than S. pasteurii. For Harbottle et al. please see: pg 116 – 117 http://orca.cf.ac.uk/108519/1/Thesis%20Stefani%20Edited.pdf. Our results indicate a much more comparable level of ureolytic activity of S. ureae compared to S. pasteurii alongside its improved cell viability during MICP treatments in sands. Harbottle et al., also found that S. ureae spores could survive long term while encapsulated in $CaCO_3$ while S. pasteurii could not. Our work, along with Harbottle et al., suggest that S. ureae has promising characteristics that may warrant future study by researchers as regards its application in ureolytic MICP.

In the revised manuscript please see: RL579-583

Comment 4 "Furthermore, the mineralogical characterizations are weak, for instance it is impossible for this reviewer to agree with the assessment of rhombohedral[ly] shaped crystals as indicated in L 414/415; also, the statement in L418 that only calcium peaks were present in the rod shaped formations cannot be followed. This reviewer clearly sees Ca, C, and O peaks in the EDS spectra in Fig 5."

We have addressed this issue in several sections of the revised manuscript and realize that the interpretation of the current data and figures as presented was incorrect.

In the revised manuscript please see: RL415-419, RL455, RL599-616

Comment 5 "The following detailed comments will support the assessment above as well as provide examples of sections and approaches, that make this manuscript hard to review and demonstrate at least some of the deficiencies that have to be remediated prior to publication."

We realize that some important information is unclear in the original text and that some information is incorrect. We thank the reviewer for such a detailed and thorough comment section. We have addressed these issues as follows with replies to the line-by-line comments that reference the relevant section(s) of the revised manuscript.

L21 units need to be defined

Please see: RL21

L 143 unclear whether the OD measurements were taken in systems in which CaCO3 precipitation occurred – if so, the authors need to explicitly discuss the possible influence of CaCO3 precipitation on OD measurements

Please see: RL128-133

Note: Where OD measurements were taken, they were from media system where no CaCO3 precipitation was expected (i.e. UB-1 and UB-2 media).

L 147 the authors assume a SG of 1 for the fluids – this might or might not be a good assumption depending on the concentrations of urea and calcium used.

Please see: RL156-158

Note: An SG of 1 was assumed only for making dilutions necessary for the ammonia-ammonium measurements. As dilutions were between 50-1000X and diluted with ddH2O this was a very good estimate relative to error that would be caused by volume delivery by pipette (which could be as high as 5%).

L 151 – unclear what HACH assay was used

Please see: RL154-156

L 163 – should this sand indeed be described as 'uniform'?

Please see: RL167

L 166 unclear how OD was measured (what was the blank, what was the pathlength – all this must be stated 189 '3 times each at 24 h periods' – unclear)

Please see: RL175 and RL128-133

L 220-224 the authors need to check whether their statements are clear and not contradictory.

Please see: RL190-195

L 230/231 the authors will have to discuss the effect of drying at 65 deg C

Please see: RL223-226

L 236 the authors should discuss to what extent a 1 month duration can be considered 'long-term' – this reviewer thinks years to decades would be considered long-term for building materials and soil stabilization

Please see: RL240

L 274 onward – as indicated above, these data are not really easily comparable since bacterial cell densities (or biomass) were not accounted for. – culture density will play an important role in the ureolytic activity

Please see: RL154-159 and RL331

L 344-346 AND L 355-357 it is unclear what is being compared and what the p-values are indicating (or not)

Please see: RL342-347, RL354-356 and RL357-360

Note: The statistical tests utilized for testing statistical significance are outlined in section 2.6 of the manuscript (RL266-271).

L 360-364 the authors should discuss why acidity might increase and discuss & compare these parameters and treatments in more detail.

Please see: RL557-560

L 371-373 the discussion in these lines is weak and not well organized. – This reviewer is unable to understand what is being discussed and compared

Please see: RL366-376

L 488 'Destruction of MICP sands'? – what does that mean?

Please see: RL491-498

L490 'increase'? compared to what?

Please see: RL491-498

L 491-493 again, this section and statistical analyses are not clear – this reviewer might be able to see that some of the specimen retained their strength once exposed to water or freezing but strength did not 'increase' – did it?

Please see: RL491-498

Note: The statistical tests utilized for testing statistical significance are outlined in section 2.6 of the manuscript (RL266-271).

L 521/522 this author agrees that monitoring ammonium concentrations is a mediocre way to assess urea hydrolysis rate since among other factors, ammonia volatilization and ammonium uptake can affect concentrations. Hence, many groups working on urea hydrolysis-induced CaCO3 precipitation are not using direct urea measurements using either spectrophotometric assays (e.g. Jung assay) or HPLC-based analyses. – The authors must discuss in more detail how the ammonium/ammonia-based estimates of urea hydrolysis rate might have affect their assessments

We agree that other robust methods for measuring MICP capability, such as measuring dissolved Ca2+ levels overtime, are also appropriate for the assessment of the ureolytic activity of an organism. However, each assay has its limitations. For example, the use of dissolved calcium ions as a measure of ureolytic activity is limited by the adsorption of calcium to solid matter (i.e. sand particles, soil, glass surfaces, etc). This technique has been used by Harbottle et al. (see reference link above – pg. 116) who has noted these limitations. Likewise, the measurement of ureolytic activity as ammonia-ammonium measurements overtime is limited too. While a quantitative urea hydrolysis rate is unable to be determined, as discussed in the manuscript the data does indicate broad bacterial activity in ammonia-ammonium production. We believe this is reasonable evidence for modest conclusions that the use of urea as a nitrogen source is medium dependent for certain strains of Bacillus but not Sporosarcina. Also, that the activity is higher in Sporosarcina then in Bacillus and that the levels of activity are similar between S. ureae and S. pasteurii. We do agree that no completely quantitative urea hydrolysis rate can be determined from the current data of this study as has been measured by other groups (see: Lauchnor et al. (2015) Journal of Applied Microbiology 118: 1321-1332.)

Please see: RL526-530

L 549-553 incomprehensible section

Please see: RL553-563

L 566 'S. urea may use the proton gradient' – unclear what 'proton gradient' – also unclear what 'may' means here

Please see: RL570-571 and RL573-577

L 572-573 what are 'co-capable candidates'?

Please see: RL581-583

L 581 why 'only' – only compared to what?

Please see: RL590

L 585 – without showing replicates and applying proper statistical tests, any statement comparing strengths of specimen will remain highly speculative

Please see: RL591-593

L 594 – still don't agree with 'rhombohedral' statement

Please see: RL602-619

L 597/598 'Media and S. subtilis treated sands gave no discernable CaCO3 formation' – where are those data? Please see: RL621

L 600 what was discussed in this sentence is not supported by Figure 4 (or is it?)

Please see: RL602-619

L 600-604 these lines make little sense to this reviewer

Please see: RL602-619

L 606-610 these lines make little sense to this reviewer

Please see: RL602-619

L 615/616 the authors should consider and discuss that dead cells can also function as nucleation points. Plus all the sand surfaces can as well. Thus, more cells might not result in a significantly increased number of nucleation points

Please see: RL632-641

L 621 'gave rise to non-significantly' – again, the statistical tests used by the authors are unclear to this reviewer and it is something different to 'not statistically significantly different' – this section once again does not make much sense to this reviewer

Please see: RL642-645

Editorial comments (not complete, just the ones that I spent the time on noting)

Urea hydrolysis equation CO(NH2)2 not CO(NH3)2

Please see: RL63

L 79 "spahericus"

Please see: RL79

L92 "alterative"

Please see: RL92

L134 "run" vs. "incubated"

Please see: RL139

L180 "fresh sample inoculate" – what is that?

Please see: RL182

L 186: 'Each mould had equipped" !?

Please see: RL188-189

L 188 'Silica' or 'Silica sand'?

Please see: RL189-195

L 228 "Visualization . . . was carried out . . .'???

Please see: RL 232

L 244 'the trials' do not 'withstand'

Please see: RL247

L 386 Fig 3 'subtilis' not 'Subtilis'

Please see: RL390

L 509: 'It was chief to understand'?

Please see: RL514

L 512: 'regardless of source nitrogen availability as yeast extract or urea'? – can't Follow

Please see: RL517

L518 one medium, two (or more) media

Please see: RL523

L 527 and 532 why 'see also'?

Please see: RL535

L 561 'Returning to s. ureae' ?? what does that mean?

Please see: RL570

L 565 'Whiffin'

Please see: RL576

L 605 'cell viability of inoculates' – what is an inoculate?

Please see: RL620

L 616-619 – language issues – this is almost incomprehensible

Please see: RL635-656

L 625/626 – language issues – this is almost incomprehensible

Please see: RL635-656

L 635 'may prove'? or may not prove . . .

Please see: RL655-657

L 639 – ' would be quite proximal'? – what does that mean?

Please see: RL658

L 647 'remain' ? – maybe 'maintain'? References have random (or not so random) spaces in them that make no sense. Please see: RL666-667

Please also note the supplement to this comment:
https://www.biogeosciences-discuss.net/bg-2017-517/bg-2017-517-AC3-supplement.pdf

**Supplement:**

[revised manuscript text omitted]

* * *
**Comment [L6]:** Editorial comment
Typographical error corrected

**Comment [L7]:** Editorial comment
Rearranged literature sources chronologically.

**Comment [L8]:** Editorial comment
Typographical error corrected

**Comment [L9]:** While not suggested, previous Genus designation stated

**Comment [L10]:** Previous Genus designation stated as suggested

(BD Bacto[TM]), Tris-Base (Trizma[TM]), 5 g/L ammonium sulfate (Molecular biology grade, Sigma-Aldrich), 10 g/L urea (Molecular biology grade, Sigma-Aldrich), pH 8.6 . The culture broth, ATCC Medium 3 (3 g/L Beef extract [BD Bacto[TM]]

115 and 5 g/L peptone [BD Bacto[TM]]) was used for *B. megaterium*, *L. sphaericus* and *B. subtilis*. and grown at 30 °C, unless otherwise specified. Colonies of *Bacillus* and *Sporosarcina* were maintained on plates prepared as described supplemented with 15 g/L agar [BD Difco[TM]] . *E. coli* was grown in Luria-Bertani (LB) broth (10 g/L tryptone [Molecular biology grade, Sigma-Aldrich], 5 g/L yeast extract [BD Bacto[TM]], 10 g/L NaCl [Molecular biology grade, Sigma-Aldrich], pH 7.5) and maintained on LB plates at 37 °C supplemented with 15 g/L agar (BD Difco[TM]). Long term stocks of all cultures were

120 prepared as described (Moore and Rene, 1975) but using dry ice as the freezing agent.

**2.2 Chemical and Biological Analysis**

**2.2.1 Culturing**

125

Single colonies were lifted and grown overnight at 200 RPM in 5mL of respective strain culture medium in a 15 mL Corning Falcon© tube. The overnight stock was combined with 200 mL of appropriate culture medium in a 500 mL Erlenmeyer flask and cultured at 175 RPM. The optical density at 600 nm ($OD_{600}$) was used to track changes in turbidity of a culture volume using a Biomate UV-Vis spectrophotometer (Thermoscientific) where 1 mL of culture volume was placed

130 into 1.5mL polystyrene cuvettes (BioRad) of a 1 cm path length. Ultra-pure water (ddH$_2$O) was used as a blank. At $OD_{600}$ values greater than 0.4, samples of culture volumes were diluted 10-100X in Tris buffered saline (TBS; 50 mM Tris-base [Trizma©, Sigma-Aldrich], 150 mM NaCl [Molecular biology grade, Sigma-Aldrich], pH 7.5) to maintain a linear relationship between turbidity and cell growth. When $OD_{600}$ reached ~ 0.5, the culture was twice spun at 5000 RPM for 5 minutes followed by a pellet re-suspension in 50 mL TBS each time. Next a fraction of volume was removed, spun at 5000

135 RPM for 5 minutes and re-suspended ($OD_{600}$ ~ 0.2) in 200 mL of a urea broth (UB) medium in a 500 mL Corning PYREX© round glass media storage bottle containing a modified Stuart's Broth (Stuart et al., 1945) as follows: 20 g/L Urea (BioReagent, Sigma-Aldrich), 5 g/L Tris-Base (Trizma©, Sigma-Aldrich), 1 g/L glucose (Reagent grade, Sigma-Aldrich), pH 8.0, with (UB-1) or without (UB-2) 10 g/L yeast extract (YE) (BD Difco[TM]). A negative control included a medium only condition. All steps were performed aseptically with preparations incubated at 150 RPM at 30 °C in triplicate for each

140 medium condition: UB-1 and UB-2. Each culture for a medium condition was staggered 10 min apart and observed for 12 h, with duplicate 2.5 mL aliquots aseptically withdrawn every 1hr, beginning at time zero (t = 0 h). The entire protocol was performed twice for a total of 6 data sets (n = 6), measured in duplicate, per culture in a single medium condition.

**2.2.2 Total Ammonia (NH$_3$-NH$_4^+$), pH and growth (OD-600) aliquots**

145

To evaluate different cell parameters efficiently, duplicate aliquots (2.5mL) were taken for tracking pH, $OD_{600}$ and NH$_3$-NH$_4^+$ production. In brief, first, whole aliquot volume pH was taken with a SB20 symphony pH probe (VWR). Next, a 1mL volume was removed for $OD_{600}$ reading as described (2.2.1). Finally, a 500 uL sample for NH$_3$-NH$_4^+$ analysis was retrieved and diluted in 500 uL of ddH$_2$O and stored as described by HACH Inc. (Hach Co. 2015) with the following additional

150 modifications made: -20 °C storage, 1 drop 5 N H$_2$SO$_4$.

**Comment [L11]:** L166 modification
Description of optical density / cellular density procedure used for all aspects of the study.
If desired the following procedural notes could also be added:

While ddH2O was used as a blank, samples diluted 10-100X had a corresponding TBS sample diluted with medium only in that range.
E.g. Sample 1-10 diluted 20X in TBS (50uL culture volume in 950uL TBS). Then, a sample diluted 20X with medium only was prepared (50mL medium in 950uL TBS). Final OD600 values reported subtracted this value from all sample values measured for a dilution.
For undiluted samples, a sample with medium only was prepared.

**Comment [L12]:** L166 modifications
Based on previous calibrations performed from recommendations in :

(1) Koch, AL. 1994. "Growth Measurement" IN: Methods for General and Molecular Bacteriology Gerhardt, P et al (ed) American Society for Microbiology, Washington, DC. p. 248-277.
(2) Collins and Lyne's Microbiological Methods" ( Estimating Microbial Numbers – p. 144-155)

**Comment [L13]:** Typographical error changed

**Comment [L14]:** L166 modification
A fraction was equal to the volume required to reach OD600 ~0.2 from the equation C1V1 = C2V2 where C1 was the OD600 of the concentrated culture in TBS, and C2 = ~0.2 , V2 = 200mL and V1 = volume to be spun down.

**Comment [L15]:** Editorial comment
Word change made 'run' to 'incubated'

**Comment [L16]:** Typographical error changed

**Comment [L17]:** L151 modification
URL provided for the downloadable technical manual of total ammonia measurement by HACH.

**Comment [L18]:** L151 modification
Sentences hereafter removed.

**2.2.3 Spectrophotometric analysis of $NH_3$-$NH_4^+$**

Samples were thawed and neutralized with 5 N NaOH as described by HACH Inc. (Hach Co. 2015). $NH_3$-$NH_4^+$ measurements were then performed as outlined (HACH Co. 2015) based on an adaptation of the work by Reardon et al. (1966) using a portable DR2700 HACH spectrophotometer. Samples were brought to a measureable range (0.01 to 0.50 mg/L $NH_3$-N) where required. Measurements for appropriate dilutions were made by mass and corrected to volume assuming a density of 1 g/L. Final values were reported as, 'U / mL' where units U = µmol of $NH_3$-$NH_4^+$ produced per minute and mL = mL solution normalized to culture density ($OD_{600}$) starting from t = 1 h.

**2.3 Microbial cementation**

**2.3.1 Model sand**

Industrial quality, pure coarse silica sand (Unimin Canada Limited) was examined with the following grain distribution where $D_{10}$, $D_{50}$, $D_{60}$ are 10 %, 50 % and 60 % of the cumulative mass: $D_{10}$ = 0.62 mm, $D_{50}$ = 0.88 mm, $D_{60}$ = 0.96 mm. The uniformity coefficient, $C_u$ was 1.55 indicating a poorly graded (i.e. uniform) sand as designated by the Unified Soil Classification System (USCS) (ASTM, 2017). A poorly graded soil was used as a model due to its undesirable geotechnical characteristics in construction (i.e., settling) and tendency for instability in nature (i.e., liquefaction) (Nakata et al., 2001; Scott, 1991).

**2.3.2 Cementation medium (CM) and culture**

Cells of each strain were grown in 1L of their respective medium split into two 1 L Erlenmeyer flasks containing 500 mL medium each at 175 RPM to an $OD_{600}$ of ~ 1.5 - 2.0 as described (2.2.1). Cells were then harvested and successively concentrated over three runs to 50 mL. Runs involved a spin down at 5000 RPM for 5 min followed by a pellet re-suspension in TBS. Prior to sand inoculation, 50 mL of a two-times (2X) concentrated cementation (CM) medium (2X CM; 0.5 M $CaCl_2$ [Anhydrous granular, Sigma-Aldrich], 0.5 M urea [BioReagent, Sigma-Aldrich], 5 g/L yeast extract [YE] [BD Difco™], 50 mM Tris-Base [Trizma©, Sigma-Aldrich], pH 8) was added to the final suspension. Negative controls were 1:1 mixes of $ddH_2O$ and 2X CM as well as the non-ureolytic strain (BGSC 3A1[T]) *B. subtilis* (Cruz-Ramos et al., 1997). A positive control with *S. pasteurii* (ATCC 11859), a ureolytic organism capable of ureolytic MICP, (van Paassen et al., 2009) was also run. The procedure was repeated every 24 h to provide fresh cells for injection during cementation trials.

**2.3.3 Sample preparation and cementation trial**

Triplicate test units were constructed from aluminum (Fig. 1), each housing a triplicate set of sample moulds measuring 60 x 60 x15 mm. Moulds were sized according to the sample intake for the direct shear apparatus (Model: ELE-26-2112/02) utilized in confined shear tests. Each mould was equipped with a drainage valve for

**Comment [L19]:** L151 modification Original literature report adapted for total ammonia analysis provided for additional clarity.

**Comment [L20]:** L147 modification Density assumption stated here only.

**Comment [L21]:** L274 modification Alternative definition: umol of NH3-NH4+ produced per minute per mL solution, normalized to cellular density (OD600) , where U / mL = umol/min/mL/OD600.

**Comment [L22]:** L163: Please see USCS designations for a sand that is 'poorly graded' (i.e. having uniform particle sizes) and 'coarse' grained (i.e. > 50% mass retained on no. 200 sieve with > 50% passing no. 4 sieve) with less than 5% fines.

See ASTM 2011.

A convenient web link:

https://en.wikipedia.org/wiki/Unified_Soil_Classification_System

**Comment [L23]:** L166 modification

**Comment [L24]:** Typographical error corrected

**Comment [L25]:** Editorial comment 'Fresh sample inoculate' to 'fresh cells'

**Comment [L26]:** Typographical error corrected

**Comment [L27]:** Editorial comment 'had equipped' to 'was equipped'

[revised manuscript text omitted]

**Comment [L33]:** L228 : Editorial comment
Replaced 'Visualization' with 'observed'

**Comment [L34]:** L236 modification.
Wording changed to reflect the incubation period.

**Comment [L35]:** Typographical change made

**Comment [L36]:** Editorial comment
Change from 'the trials' and 'withstand' to 'treated silica sand' and 'impacted'.

**2.6 Statistical processing**

265

All statistical manipulations were performed in Excel (2007). Sample means were reported alongside the standard error of the mean (SE) or standard deviation (SD). Normality of all data sets were confirmed with the Anderson-Darling test ($\alpha = 0.05$). The Student's t-test (unpaired, two-tailed; $\alpha = 0.05$) was utilized to compare sample means of experimental conditions for statistical significance. Prior to each t-test, homogeneity of variances for data sets were

270 determined using a F-test ($\alpha = 0.05$). Where variances were statistically observed as unequal, a Welch's t-test was adapted to test statistical significance between two sample means.

**3 Results**

275 **3.1 $NH_3$-$NH_4^+$ production**

Among the different bacterial strains considered, *S. pasteurii* (32.50 U/mL [UB-1]; 32.76 U/mL [UB-2]) *and S. ureae* (29.00 U/mL [UB-1]; 30.28 U/mL [UB-2]) were capable of producing the first and second highest levels of $NH_3$-$NH_4^+$, respectively, per unit of time, in both UB-1 and UB-2 medium (Fig. 2). Isolates of *B. subtilis* (2.91

280 U/mL), *B. megaterium* (4.87 U/mL) and *L. sphaericus* (5.89 U/mL) displayed a lower peak of $NH_3$-$NH_4^+$ production in both media. When urea in medium moved from the sole source (i.e., UB-2) to one of a number of sources (i.e., UB-1) for nitrogen, $NH_3$-$NH_4^+$ production dropped to near zero values (Fig. 2) for *B. subtilis* (0.44 U/mL), *B. megaterium* (0.56 U/mL) and *L. sphaericus* (1.20 U/mL) that were statistically significantly different ($p < 0.05$, $n = 6$) from the final UB-1 values for each species. However, isolates of *S. ureae* and *S. pasteurii* observed no statistically significant

285 difference ($p > 0.05$, $n = 6$) between final values recorded in UB-1 and UB-2 medium. Instead, a rise in production ($t = 0 – 5$ h) followed by a levelling off in value ($t = 6 – 12$ h) was the general trend observed in UB-1 and UB-2 medium (Fig. 2).

**Comment [L37]:** Typographical change made

**Comment [L38]:** Modifications made to clarify that urea was the only source of nitrogen in UB-2 while one of a number of sources for nitrogen in UB-1.

**Comment [L39]:** 'significant' reworded to 'statistically different'

**Comment [L40]:** 'non significant' reworded to 'no statistically significant difference'

**Comment [L41]:** Typographical change made

[revised manuscript text omitted]

**Comment [L43]:** Typographical error correction

**Comment [L44]:** L344-346 correction

**Comment [L45]:** Wording modified to reflect that differences were statistically different and decreased. Also reference section 2.6 for specific tests used, where appropriate.

**Comment [L46]:** Additional observations noted for growth in medium UB-1 and UB-2

**Comment [L47]:** L355-357 correction

**Comment [L48]:** Additional details provided on statistical data used (also reference section 2.6 for specific tests used, where appropriate)

**Comment [L49]:** Additional details provided on statistical data used (also reference section 2.6 for specific tests used, where appropriate)

**Comment [L50]:** Additional details provided on statistical data used (also reference section 2.6 for specific tests used, where appropriate)

375 *ureae*], -75.4 % [*S. pasteurii*], -77.7 % [*B. subtilis*]) were not statistically significantly different ($p > 0.05$, $n = 9$) when comparing values between species. Of note, the medium-only control observed no colony growth before and after incubation.

**Comment [L51]:** L371-373 corrections

[Figure]

**Comment [L52]:** L386: 'Subtilis' to 'subtilis'

[Figure]

Fig. 3. Direct shear strengths ($\tau$, *kPa*) of treated sands (*SE, n = 3*).

[Figure]

Fig. 4. Microbial viability of treated sands before injection (*black bars*) and after incubation (*gray bars*) (*SD, n = 9*).

**3.5 Microstructure investigation**

415 The precipitation of calcium as $CaCO_3$ via MICP was visualized. Sand granules from approximately the first 1cm of sands treated with MICP solution (i.e., CM-1) combined with *S. ureae* are shown (Fig. 5) where crystals arranged in rosette peaks (20 – 40 μm) can be seen across the surface of a sand grain (Fig. 5). Rod-shaped structures (40 – 80 μm) can also be visualized, though less commonly, across grain surfaces (Fig. 5). Calcium, carbon and oxygen peaks captured by EDS analysis for crystals organized in 'rosette' patterns as well as in rod-shaped structures suggest $CaCO_3$ precipitation.

420

**Comment [L53]:** Additional corrections made in line with L414/415, L418 and L594 comment

425

430

435

440

445

(a)

450

455

460

**Comment [L54]:** Arrows rearranged in image (a) to match magnified position in image (b)

(b)

SEI 5kV    x1,100    10µm

**(c)**                    **(d)**

[revised manuscript text omitted]

**Comment [L63]:** Changed 'co-nitrogen' to 'alternative nitrogen'.

**Comment [L64]:** L 549-553 correction

**Comment [L65]:** Phrase reworded to better clarify other factors contributing to a decreased growth rate overtime.

**Comment [L66]:** L360-L364 : Discussion of why acidity in UB-1 medium would increase over time included.

**Comment [L67]:** Typographical error; changed 'media' to 'medium'

**Comment [L68]:** L561 : Removed 'returning to S. ureae'. Started new paragraph.

**Comment [L69]:** Word change; 'high' to 'ureolytic'

**Comment [L70]:** Typographical error; change 'alkalophile' to 'alkaliphile'

**Comment [L71]:** L565; World change, 'Whiffen' to 'Whiffin'

**Comment [L72]:** L566 correction

**Comment [L73]:** L572-573 ; Included work of Harbottle et al. (2016) as well as Sarmast et al. (2014) to provide context for this work's conclusions in relation to those found by other groups working with S. ureae on MICP.

[revised manuscript text omitted]

Comment [L74]: L581 : Removed 'only'

Comment [L75]: L585 : Changed to better explain improvements in strength of S. ureae treated samples in relation to the control treatments.

Comment [L76]: Additional details provided on statistical data used (also reference section 2.6 for specific tests used, where appropriate)

Comment [L77]: Typographical error change; 'an' to 'a'

Comment [L78]: Typographical error change; 'sp.' to 'species'

Comment [L79]: Changed to clarify that evidence supports S. ureae can improve soils by ureolytic MICP, much like S. pasteurii

Comment [L80]: L594 ; 'Rhombohedra' changed to more qualitative descriptions based on the SEM images provided.

Comment [L81]: L594, L600, L600-604: This section is rewritten to better reflect the mineralogy and microscopy techniques used to identify mineral precipitation along MICP treated surfaces.

Comment [L82]: Typographical error ; 'media' to 'medium'.

Comment [L83]: L597/98: Data not shown but available upon request.

Comment [L84]: L594, L600, L600-604 : This section is added to give context for why SEM analysis was performed.

Comment [L85]: Typographical error ; 'analyzed' to 'analysed'

Comment [L86]: L606-610 ; This section is reworded to better clarify that differences in the cell counts for S. ureae / S. pasteurii compared to B. subtilis could be due to the dilution medium used for cell counts.

[revised manuscript text omitted]

---

## Author Response (AR1)

**Author's Response – Editor**

**(1)    Author's Response**

Dear Dr. Akob,

We thank you for your time reviewing our manuscript and making insightful editorial comments for content changes to be made. We have considered your changes and have now made the appropriate additions to the manuscript.

For your convenience, we have highlighted the sections in the manuscript in yellow which reflect these changes alongside comment bubbles, where appropriate, which provide additional rationale / clarification for a change.

We do welcome any further comments regarding how we may make the manuscript stronger and, specifically, if the above rationale can be made more clearly in the current manuscript.

Thank you again for your time and kind efforts providing your editorial expertise.

**(2)    Author's Changes in Manuscript**

1.  L. 21: can you define "U" as units here?

    Done

2.  L. 139: use subscript for $OD_{600}$

    Modified

3.  L. 178: provide the strain designation after the species name

    Provided

4.  L. 195: Sporosarcina is spelled wrong.

    Edited

5.  When referring to figures please give the panel letter in the text (e.g., l. 276, l. 279 and elsewhere)

    Panels letters have now been added. Please see yellow highlighted texts for the appropriate additions made throughout the manuscript.

6. L. 274: the inclusion of parenthetical information here is a bit awkward. Consider revising.

   Revised

7. L. 228-231: did you have any other SEM prep steps? Did you dehydrate the samples or mount them onto anything?

   Details on mounting steps added.

8. L. 339: since you are working with liquid cultures colony is not correct. Modify

   Modified to 'cell abundance'

9. Figure 2: it is difficult to tell the circles from the squares because of the error bars, consider modifying with color or gray scale. It would be more logical to put UB-1 medium on the left and UB-2 on the right. Also, define YE in the legend and specify that it's a N source.

   All suggestions made and detailed in Fig. 2 as requested.

10. L. 368-369: revise to "media (i.e., CM-1) significantly (p < 0.05) improved in shear strength compared to control vessels (15.77 kPa) with MICP media only"

    Revised as suggested with some modifications as per comments from Editor #3.

11. L. 509-576: please break this up into more than 1 paragraph

    Divided into 3 distinct pargraphs as detailed below (in yellow)

12. L. 613: not significantly

    Change made

13. L. 632 and elsewhere: it seems that this should be abundance and not "colony total"

    Modified as detailed above for L339 comment.

14. L. 646: do you mean retain?

    Changed to clarify 'retention' of direct shear strength

15. L. 652: do you mean "reflective of"?

    Editorial change made.

**Improving the Strength of Sandy Soils via Ureolytic CaCO$_3$ Solidification by *Sporosarcina ureae**

Justin Michael Whitaker[1], Sai Vanapalli[2], and Danielle Fortin[1];

[1]Department of Earth and Environmental Sciences (413-ARC). University of Ottawa, K1N 6N5, Ottawa, ON, Canada
[2]Department of Civil Engineering (A015-CBY). University of Ottawa, K1N 6N5, Ottawa, ON, Canada

*Correspondence to:* D. Fortin (dfortin@uottawa.ca)

Key words: Urease, calcite precipitation, MICP, *Sporosarcina, Bacillus,* biomineralisation, biofilm

**Comment [L1]:** Typographical error corrected

**Abstract**

'Microbial induced carbonate precipitation' (MICP) is a biogeochemical process that can be applied to strengthen materials. The hydrolysis of urea by microbial catalysis to form carbonate is a commonly studied example of MICP. In this study, *Sporosarcina ureae*, a ureolytic organism, was compared to other ureolytic and non-ureolytic organisms of *Bacillus* and *Sporosarcina* in the assessment of its ability to produce carbonates by ureolytic MICP for ground reinforcement. It was found that *S. ureae* grew optimally in alkaline (pH ~9.0) conditions which favoured MICP and could degrade urea (units [U] /mL = $\mu$mol/min.mL.OD$_{600}$) at levels (30.28 U/mL) similar to *S. pasteurii* (32.76 U/mL), the model ureolytic MICP organism. When cells of S. ureae were concentrated (OD$_{600}$ ~15-20) and mixed with cementation medium containing 0.5 M calcium chloride (CaCl$_2$) and urea into a model sand, repeated treatments (3 x 24 h) were able to improve the confined direct shear strength of samples from 15.77 kPa to as much as 135.80 kPa. This was more than any other organism observed in the study. Imaging of the reinforced samples with scanning electron microscopy and energy dispersive spectroscopy confirmed the successful precipitation of calcium carbonate (CaCO$_3$), across sand particles by *S. ureae*. Treated samples were also tested experimentally according to model North American climatic conditions to understand the environmental durability of MICP. No statistically significant ($p < 0.05$, n = 3) difference in strength was observed for samples that underwent freeze-thaw cycling or flood-like simulations. However, shear strength of samples following acid-rain simulations fell to 29.2% of control MICP samples. Overall, the species *S. ureae* was found to be an excellent organism for MICP by ureolysis to achieve ground strengthening. However, the feasibility of MICP as a durable reinforcement technique is limited by specific climate conditions (i.e. acid rain).

**Comment [L2]:** L21 modification Change from U / mL to units [U] / mL

**Comment [L3]:** L21 modification Alternative units: umol/min.mL.OD600

**Comment [L4]:** Typographical error corrected

**Comment [L5]:** Clarified sample size (n) and wording to specify a statistical test was performed (as described in section 2.6)

[revised manuscript text omitted]

**Comment [L8]:** Editorial comment
Typographical error corrected

**Comment [L9]:** Editorial comment
Rearranged literature sources chronologically.

**Comment [L10]:** Editorial comment
Typographical error corrected

**Comment [L11]:** While not suggested, previous Genus designation stated

**Comment [L12]:** Previous Genus designation stated as suggested

(BD Bacto[TM]), Tris-Base (Trizma[TM]), 5 g/L ammonium sulfate (Molecular biology grade, Sigma-Aldrich), 10 g/L urea (Molecular biology grade, Sigma-Aldrich), pH 8.6 . The culture broth, ATCC Medium 3 (3 g/L Beef extract [BD Bacto[TM]]

205 and 5 g/L peptone [BD Bacto[TM]]) was used for *B. megaterium*, *L. sphaericus* and *B. subtilis*. and grown at 30 °C, unless otherwise specified. Colonies of *Bacillus* and *Sporosarcina* were maintained on plates prepared as described supplemented with 15 g/L agar [BD Difco[TM]] . *E. coli* was grown in Luria-Bertani (LB) broth (10 g/L tryptone [Molecular biology grade, Sigma-Aldrich], 5 g/L yeast extract [BD Bacto[TM]], 10 g/L NaCl [Molecular biology grade, Sigma-Aldrich], pH 7.5) and maintained on LB plates at 37 °C supplemented with 15 g/L agar (BD Difco[TM]). Long term stocks of all cultures were

210 prepared as described (Moore and Rene, 1975) but using dry ice as the freezing agent.

**2.2 Chemical and Biological Analysis**

**2.2.1 Culturing**

215

Single colonies were lifted and grown overnight at 200 RPM in 5mL of respective strain culture medium in a 15 mL Corning Falcon© tube.  The overnight stock was combined with 200 mL of appropriate culture medium in a 500 mL Erlenmeyer flask and cultured at 175 RPM.  The optical density at 600 nm ($OD_{600}$) was used to track changes in turbidity of a culture volume using a Biomate UV-Vis spectrophotometer (Thermoscientific) where 1 mL of culture volume was placed

220 into 1.5mL polystyrene cuvettes (BioRad) of a 1 cm path length. Ultra-pure water (ddH$_2$O) was used as a blank. At $OD_{600}$ values greater than 0.4, samples of culture volumes were diluted 10-100X in Tris buffered saline (TBS; 50 mM Tris-base [Trizma©, Sigma-Aldrich], 150 mM NaCl [Molecular biology grade, Sigma-Aldrich], pH 7.5) to maintain a linear relationship between turbidity and cell growth.   When $OD_{600}$ reached ~ 0.5, the culture was twice spun at 5000 RPM for 5 minutes followed by a pellet re-suspension in 50 mL TBS each time. Next a fraction of volume was removed, spun at 5000

225 RPM for 5 minutes and re-suspended ($OD_{600}$ ~ 0.2) in 200 mL of a urea broth (UB) medium in a 500 mL Corning PYREX© round glass media storage bottle containing a modified Stuart's Broth (Stuart et al., 1945) as follows: 20 g/L Urea (BioReagent, Sigma-Aldrich), 5 g/L Tris-Base (Trizma©, Sigma-Aldrich), 1 g/L glucose (Reagent grade, Sigma-Aldrich), pH 8.0, with (UB-1) or without (UB-2) 10 g/L yeast extract (YE)  (BD Difco[TM]).  A negative control included a medium only condition. All steps were performed aseptically with preparations incubated at 150 RPM at 30 °C in triplicate for each

230 medium condition: UB-1 and UB-2.  Each culture for a medium condition was staggered 10 min apart and observed for 12 h, with duplicate 2.5 mL aliquots aseptically withdrawn every 1hr, beginning at time zero (t = 0 h). The entire protocol was performed twice for a total of 6 data sets (n = 6), measured in duplicate, per culture in a single medium condition.

**2.2.2 Total Ammonia (NH$_3$-NH$_4^+$), pH and growth ($OD_{600}$) aliquots**

235

To evaluate different cell parameters efficiently, duplicate aliquots (2.5mL) were taken for tracking pH, $OD_{600}$ and NH$_3$-NH$_4^+$ production. In brief, first, whole aliquot volume pH was taken with a SB20 symphony pH probe (VWR). Next, a 1mL volume was removed for $OD_{600}$ reading as described (2.2.1). Finally, a 500 uL sample for NH$_3$-NH$_4^+$ analysis was retrieved and diluted in 500 uL of ddH$_2$O and stored as described by HACH Inc. (Hach Co. 2015) with the following additional

240 modifications made: -20 °C storage, 1 drop 5 N H$_2$SO$_4$.

**Comment [L13]:** L166 modification
Description of optical density / cellular density procedure used for all aspects of the study.
If desired the following procedural notes could also be added:

While ddH2O was used as a blank, samples diluted 10-100X had a corresponding TBS sample diluted with medium only in that range.
E.g. Sample 1-10 diluted 20X in TBS (50uL culture volume in 950uL TBS). Then, a sample diluted 20X with medium only was prepared (50mL medium in 950uL TBS).  Final OD600 values reported subtracted this value from all sample values measured for a dilution.
For undiluted samples, a sample with medium only was prepared.

**Comment [L14]:** L166 modifications
Based on previous calibrations performed from recommendations in :

(1) Koch, AL. 1994. "Growth Measurement" IN: Methods for General and Molecular Bacteriology Gerhardt, P et al (ed) American Society for Microbiology, Washington, DC. p. 248-277.
(2) Collins and Lyne's Microbiological Methods" ( Estimating Microbial Numbers – p. 144-155)

**Comment [L15]:** Typographical error changed

**Comment [L16]:** L166 modification
A fraction was equal to the volume required to reach OD600 ~0.2 from the equation C1V1 = C2V2 where C1 was the OD600 of the concentrated culture in TBS, and C2 = ~0.2 , V2 = 200mL and V1 = volume to be spun down.

**Comment [L17]:** Editorial comment
Word change made 'run' to 'incubated'

**Comment [L18]:** Typographical error changed

**Comment [L19]:** L139: Change to subscript for OD600.

**Comment [L20]:** L151 modification
URL provided for the downloadable technical manual of total ammonia measurement by HACH.

**Comment [L21]:** L151 modification
Sentences hereafter removed.

**2.2.3 Spectrophotometric analysis of $NH_3$-$NH_4^+$**

Samples were thawed and neutralized with 5 N NaOH as described by HACH Inc. (Hach Co. 2015). $NH_3$-$NH_4^+$ measurements were then performed as outlined (HACH Co. 2015) based on an adaptation of the work by Reardon et al. (1966) using a portable DR2700 HACH spectrophotometer. Samples were brought to a measureable range (0.01 to 0.50 mg/L $NH_3$-N) where required. Measurements for appropriate dilutions were made by mass and corrected to volume assuming a density of 1 g/L. Final values were reported as, 'U/mL' where units U = µmol of $NH_3$-$NH_4^+$ produced per minute and mL = mL solution normalized to culture density ($OD_{600}$) starting from t = 1 h.

**Comment [L22]:** L151 modification
Original literature report adapted for total ammonia analysis provided for additional clarity.

**Comment [L23]:** L147 modification
Density assumption stated here only.

**Comment [L24]:** L274 modification
Alternative definition: umol of NH3-NH4+ produced per minute per mL solution, normalized to cellular density (OD600) , where U / mL = umol/min/mL/OD600.

**2.3 Microbial cementation**

**2.3.1 Model sand**

Industrial quality, pure coarse silica sand (Unimin Canada Limited) was examined with the following grain distribution where $D_{10}$, $D_{50}$, $D_{60}$ are 10 %, 50 % and 60 % of the cumulative mass: $D_{10}$ = 0.62 mm, $D_{50}$ = 0.88 mm, $D_{60}$ = 0.96 mm. The uniformity coefficient, $C_u$ was 1.55 indicating a poorly graded (i.e. uniform) sand as designated by the Unified Soil Classification System (USCS) (ASTM, 2017). A poorly graded soil was used as a model due to its undesirable geotechnical characteristics in construction (i.e., settling) and tendency for instability in nature (i.e., liquefaction) (Nakata et al., 2001; Scott, 1991).

**Comment [L25]:** L163: Please see USCS designations for a sand that is 'poorly graded' (i.e. having uniform particle sizes) and 'coarse' grained (i.e. > 50% mass retained on no. 200 sieve with > 50% passing no. 4 sieve) with less than 5% fines.

See ASTM 2011.

A convenient web link:

https://en.wikipedia.org/wiki/Unified_Soil_Classification_System

**2.3.2 Cementation medium (CM) and culture**

[revised manuscript text omitted]

Comment [L38]: L220-224
The washing, drying and testing methods are described and made more clear
L230/231 modification
The effect of drying is discussed

Comment [L39]: L228-231: Sample mounting details added. No dehydration performed but cleaning with ddH2O and drying by oven was done (as described)/

Comment [L40]: L228 : Editorial comment
Replaced 'Visualization' with 'observed'

Comment [L41]: Typographical error ; Changed from 'analyzed' to 'analysed'

Comment [L42]: L236 modification.
Wording changed to reflect the incubation period.

Comment [L43]: Typographical change made

Comment [L44]: Editorial comment
Change from 'the trials' and 'withstand' to 'treated silica sand' and 'impacted'.

**2.6 Statistical processing**

All statistical manipulations were performed in Excel (2007). Sample means were reported alongside the standard error of the mean (SE) or standard deviation (SD). Normality of all data sets were confirmed with the Anderson-Darling test ($\alpha = 0.05$). The Student's t-test (unpaired, two-tailed; $\alpha = 0.05$) was utilized to compare sample means of experimental conditions for statistical significance. Prior to each t-test, homogeneity of variances for data sets were determined using a F-test ($\alpha = 0.05$). Where variances were statistically observed as unequal, a Welch's t-test was adapted to test statistical significance between two sample means.

**3 Results**

**3.1 $NH_3$-$NH_4^+$ production**

Among the different bacterial strains considered, *S. pasteurii and S. ureae* were capable of producing the first and second highest levels of $NH_3$-$NH_4^+$, respectively, per unit of time, in both UB-1 (32.50 U/mL ; 29.00 U/mL) and UB-2 (32.76 U/mL ; 30.28 U/mL medium (Fig. 2a, 2b). Isolates of *B. subtilis* (2.91 U/mL), *B. megaterium* (4.87 U/mL) and *L. sphaericus* (5.89 U/mL) displayed a lower peak of $NH_3$-$NH_4^+$ production in both media. When urea in medium moved from the sole source (i.e., UB-2) to one of a number of sources (i.e., UB-1) for nitrogen, $NH_3$-$NH_4^+$ production dropped to near zero values (Fig. 2a, 2b) for *B. subtilis* (0.44 U/mL), *B. megaterium* (0.56 U/mL) and *L. sphaericus* (1.20 U/mL) that were statistically significantly different ($p < 0.05$, n = 6) from the final UB-1 values for each species. However, isolates of *S. ureae* and *S. pasteurii* observed no statistically significant difference ($p > 0.05$, n = 6) between final values recorded in UB-1 and UB-2 medium. Instead, a rise in production (t = 0 – 5 h) followed by a levelling off in value (t = 6 – 12 h) was the general trend observed in UB-1 and UB-2 medium (Fig. 2a, 2b).

**Comment [L45]:** L274 : Revised to list production values (U/mL) at end of sentence. L276: Revised to include panel number

**Comment [L46]:** Typographical change made

**Comment [L47]:** Modifications made to clarify that urea was the only source of nitrogen in UB-2 while one of a number of sources for nitrogen in UB-1.

**Comment [L48]:** L279: Revised to include panel number

**Comment [L49]:** 'significant' reworded to 'statistically different'

**Comment [L50]:** 'non significant' reworded to 'no statistically significant difference'

**Comment [L51]:** Typographical change made

**Comment [L52]:** Revised to include panel number

390

395

400

405

410

415

420

[Figure]

Fig. 2. **(a) , (b)** NH₃-NH₄⁺ production (U/mL = umol of $NH_3$-$NH_4^+$ /  minute.mL.OD₆₀₀ of culture) ; **(c), (d)** pH ; and **(e), (f)** growth of selected bacteria types in  **(a) , (c) , (e)** UB-1 (*No yeast extract [YE]*) and **(b) , (d) , (f)** UB-2 (*10 g/L YE*) nutrient conditions (*SD, n = 6*). YE was a nitrogen source in the growth medium.

Comment [L53]: Additional suggested change based on L274 comment

Comment [L54]: Figure 2:
(1) YE defined as yeast extract and clarified to be a N source in medium.
(2) Change from black to grey scale to better differentiate error bars from data points.
(3) Placement of UB-1 medium panels on left and UB-2 medium panels on right.

425

**3.2 Examination of colony abundance in culture**

All strains showed a decline in growth progression when medium was restricted (i.e., UB-2) to urea as nitrogen and glucose as carbon, sources, respectively (Fig. 2e, 2f). Growth repression was greatest in the cases of *B. subtilis* (-33.9 %), *L. sphaericus* (-26.8 %) and *B. megaterium* (-23.6 %) compared to *S. pasteurii* (-17.8 %) and *S. ureae* (-16.6 %). Additionally, the final $OD_{600}$ (t = 12 h) achieved for all strains in UB-2 medium was decreased compared to UB-1 medium values (t = 12 h) and the difference in value for each strain was found to be statistically significantly different (p < 0.05, n = 6). Growth cessation (i.e. stationary phase) occurred for *S. ureae* and *S. pasteurii* in both conditions but later in UB-1 (t = 11 h) compared to UB-2 (t = 9 – 10 h) medium (Fig. 2e, 2f); they grew logistically in both medium conditions. In general, growth of *L. sphaericus*, *B. subtilis* and *B. megaterium* in UB-2 medium followed a logistic growth curve too. However, in UB-1 medium their growth fit an exponential model, whereby an exponential growth phase was observed from t = 4 – 12 h following a lag phase of growth between t = 0 – 3 h.

**3.3 Changes in pH**

The alkalinity increased with the increase in time for the strains of *S. ureae* and *S. pasteurii* studied, in both UB-1 (8.99, 9.2) and UB-2 (8.74, 8.8) medium. The lowest final pH values were observed in *L. sphaericus* (7.88; 8.16), *B. megaterium* (7.85 ; 7.93) and *B. subtilis* (7.70 ; 7.81) in UB-1 and UB-2 medium, at the end of 12 h (Fig. 2c, 2d). While pH continued to rise for *S. pasteurii* and *S. ureae* in either UB-1 or UB-2 medium, it was constant for *L. sphaericus*, *B. megaterium* and *B. subtilis* after time in UB-1 medium as early as 6 h (*L. sphaericus* and *B. megaterium*) in UB-2 medium. While final pH values for *L. sphaericus*, *B. megaterium* and *B. subtilis* reached higher final (t = 12 h) values in UB-2 medium compared to UB-1, that were found to be statistically significantly different (p < 0.05, n = 6), the opposite was true for *S. pasteurii* and *S. ureae*; values in UB-2 were lower compared to UB-1 and the difference was found to be statistically significantly different for each species (p < 0.05, n = 6). In general, acidity increased with the increase in time for *L. sphaericus*, *B. megaterium* and *B. subtilis* in UB-1 medium. This was also true in UB-2 medium except for *L. sphaericus* which showed an increase in pH over time.

**3.4 Mechanical and biological behaviour in MICP reinforced sands**

Experiments of sand consolidation with triplicate holding vessels (Fig. 1) mixed with *S. ureae* (135.77 kPa) or *S. pasteurii* (135.5kPa) and fed MICP medium (i.e., CM-1) had improvements in their direct shear strength compared to control vessels (15.77 kPa) fed with MICP medium only. In fact, the difference in direct shear strength values for *S. ureae* and *S. pasteurii* compared to control vessels were found to be statistically significantly different (p < 0.05, n = 3). However, the difference in strength between *S. ureae* and *S. pasteurii* were not statistically significantly different (p > 0.05, n = 3). Mixtures of non-ureolytic *B. subtilis* (28.1 kPa) showed no statistically significant difference (p > 0.05, n = 3) in value when compared to the control (Fig. 3). While pre-injection (21.9 x $10^7$ CFU/mL) and post incubation (3.2 x $10^7$ CFU/mL) cell abundance was highest in the case of *B. subtilis*, (Fig. 4) all bacterial isolates showed a decrease in cell abundance when comparing pre-injection to post incubation cell abundance with statistically significant differences (p < 0.05, n = 9). Also, the percentage loss of cell abundance, taken as the difference between post incubation and pre-

Comment [L55]: L339: Changed from 'colony total' to 'colony abundance'

Comment [L56]: Revised to include panel number

Comment [L57]: Typographical error correction

Comment [L58]: L344-346 correction

Comment [L59]: Wording modified to reflect that differences were statistically different and decreased. Also reference section 2.6 for specific tests used, where appropriate.

Comment [L60]: Revised to include panel number

Comment [L61]: Additional observations noted for growth in medium UB-1 and UB-2

Comment [L62]: Revised to include panel number

Comment [L63]: L355-357 correction

Comment [L64]: Additional details provided on statistical data used (also reference section 2.6 for specific tests used, where appropriate)

Comment [L65]: Additional details provided on statistical data used (also reference section 2.6 for specific tests used, where appropriate)

Comment [L66]: L368-369: Suggested changes made.

Comment [L67]: Additional details provided on statistical data used (also reference section 2.6 for specific tests used, where appropriate)

Comment [L68]: Change from 'colony total' to 'cell abundance

465    injection cell abundances divided by the initial pre-injection cell abundance (-77.7 % [*S. ureae*], -75.4 % [*S. pasteurii*], -

77.7 % [*B. subtilis*]) were not statistically significantly different (p > 0.05, n = 9) when comparing values between

species.  Of note, the medium-only control had no cell growth (CFU/mL) observed before and after incubation.

**Comment [L69]:** L371-373 corrections

**Comment [L70]:** Change from 'colony growth' to 'cell growth'

[Figure]

[Figure]

**Comment [L71]:** L386: 'Subtilis' to 'subtilis'

470

475

480    Fig. 3. Direct shear strengths ($\tau$, *kPa*) of treated sands (*SE, n = 3*).

[Figure]

485

490

495

500    Fig. 4. Microbial viability of treated sands before injection (*black bars*) and after incubation (*gray bars*) (*SD, n = 9*).

**3.5 Microstructure investigation**

505 The precipitation of calcium as $CaCO_3$ via MICP was visualized. Sand granules from approximately the first 1cm of sands treated with MICP solution (i.e., CM-1) combined with *S. ureae* are shown (Fig. 5a, 5b) where crystals arranged in rosette peaks (20 – 40 μm) can be seen across the surface of a sand grain (Fig. 5a, 5b). Rod-shaped structures (40 – 80 μm) can also be visualized, though less commonly, across grain surfaces (Fig. 5a, 5b). Calcium, carbon and oxygen peaks captured by EDS analysis for crystals organized in 'rosette' patterns as well as in rod-shaped structures suggest $CaCO_3$

510 precipitation (Fig. 5c, 5d).

**Comment [L72]:** Revised to include panel number

**Comment [L73]:** Additional corrections made in line with L414/415, L418 and L594 comment

515

520

525

530

535

(a)                                                            (b)

[Figure]

**Comment [L74]:** Arrows rearranged in image (a) to match magnified position in image (b)

[revised manuscript text omitted]

**Comment [L77]:** L512 : Changed to specify that the nutrient medium, with or without yeast extract did not affect the ability for S. ureae to produce NH3-NH4+ from compounds like urea.

**Comment [L78]:** Revised to include panel number

**Comment [L79]:** L518: change of 'mediums' to 'media'

**Comment [L80]:** L521/522 : Additions addressing the limitations of the ammonium/ammonia measurement on the conclusions able to be drawn in the study.

**Comment [L81]:** L509-576: New paragraph created.

**Comment [L82]:** L527 – removed 'see also'

**Comment [L83]:** L532 – removed 'see also'

**Comment [L84]:** L509-576: Sentence reworded to make a new paragraph.

**Comment [L85]:** Revised to include panel number

[revised manuscript text omitted]

**Comment [L86]:** Removed 'see also'

**Comment [L87]:** Typographical error ; change from 'media' to 'medium'

**Comment [L88]:** Typographical error ; addition of 'medium'

**Comment [L89]:** Changed 'co-nitrogen' to 'alternative nitrogen'.

**Comment [L90]:** Revised to include panel number

**Comment [L91]:** Revised to include panel number

**Comment [L92]:** L 549-553 correction

**Comment [L93]:** Revised to include panel number

**Comment [L94]:** Phrase reworded to better clarify other factors contributing to a decreased growth rate overtime.

**Comment [L95]:** L360-L364 : Discussion of why acidity in UB-1 medium would increase over time included.

**Comment [L96]:** Revised to include panel number

**Comment [L97]:** Typographical error; changed 'media' to 'medium'

**Comment [L98]:** L561 : Removed 'returning to S. ureae'. Started new paragraph.

**Comment [L99]:** Revised to include panel number

**Comment [L100]:** Word change; 'high' to 'ureolytic'

**Comment [L101]:** Typographical error; change 'alkalophile' to 'alkaliphile'

**Comment [L102]:** L565; World change, 'Whiffen' to 'Whiffin'

**Comment [L103]:** L566 correction

[revised manuscript text omitted]

**Comment [L104]:** L572-573 ; Included work of Harbottle et al. (2016) as well as Sarmast et al. (2014) to provide context for this work's conclusions in relation to those found by other groups working with S. ureae on MICP.

**Comment [L105]:** L581 : Removed 'only'

**Comment [L106]:** L585 : Changed to better explain improvements in strength of S. ureae treated samples in relation to the control treatments.

**Comment [L107]:** Additional details provided on statistical data used (also reference section 2.6 for specific tests used, where appropriate)

**Comment [L108]:** Typographical error change; 'an' to 'a'

**Comment [L109]:** Typographical error change; 'sp.' to 'species'

**Comment [L110]:** Changed to clarify that evidence supports S. ureae can improve soils by ureolytic MICP, much like S. pasteurii

**Comment [L111]:** L594 ; 'Rhombohedra' changed to more qualitative descriptions based on the SEM images provided.

**Comment [L112]:** Revised to include panel number

**Comment [L113]:** Revised to include panel number

**Comment [L114]:** Revised to include panel number

**Comment [L115]:** L594, L600, L600-604: This section is rewritten to better reflect the mineralogy and microscopy techniques used to identify mineral precipitation along MICP treated surfaces.

**Comment [L116]:** Typographical error ; 'media' to 'medium'.

**Comment [L117]:** L597/98: Data not shown but available upon request.

[revised manuscript text omitted]

**Comment [L118]:** L594, L600, L600-604 : This section is added to give context for why SEM analysis was performed.

**Comment [L119]:** Typographical error ; 'analyzed' to 'analysed'

**Comment [L120]:** Change from 'colony total' to 'cell abundance

**Comment [L121]:** Change from 'colony total' to 'cell abundance

**Comment [L122]:** Change from 'colony total' to 'cell abundance

**Comment [L123]:** Changed from 'colony count' to 'serial dilution of the growth medium'

**Comment [L124]:** L606-610 ; This section is reworded to better clarify that differences in the cell counts for S. ureae / S. pasteurii compared to B. subtilis could be due to the dilution medium used for cell counts.

**Comment [L125]:** Typographical error changed ; ',.' to '.,'

**Comment [L126]:** Change from 'colony total' to 'cell abundance

**Comment [L127]:** Change from 'colony total' to 'cell abundance

**Comment [L128]:** L613: Phrase reworded to remove 'non-significant' statement.

**Comment [L129]:** Typographical error change ; change from 'bacterial' to 'bacteria'

**Comment [L130]:** Change from 'cell number' to 'cell abundance

**Comment [L131]:** Typographical error changed : 'modeled' to "modelled"

**Comment [L132]:** Change from 'cell number' to 'cell abundance

**Comment [L133]:** L632: Change from 'colony total' or 'cell number' to 'colony abundance' or 'cell abundance'.

**Comment [L134]:** Change from 'cell number' to 'cell abundance

**Comment [L135]:** Change from 'colony total' or 'cell number' to 'colony abundance' or 'cell abundance'.

740 sub-optimal spreading mechanism could have hindered strength achievement in some moulds of *S. ureae* treatment where pore plugging by organic matter (i.e., cells) occurred. This in mind, optimization of treatment protocols would help to determine whether or not *S. ureae* is the superior candidate compared to *S. pasteurii* given that it has consistently increased total cell abundance (Fig. 3) to support more nucleation of $CaCO_3$ overtime, in tandem with a $NH_3$-$NH_4^+$ production comparable to that of *S. pasteurii*. However, it is important to note that *S. ureae* cells are significantly

745 smaller than cells of *S. pasteurii* (Claus and Fahmy, 1986). Therefore, the total cellular surface area available for nucleation of $CaCO_3$ would be similar for the two species. This provides a possible explanation for why no statistically significant differences in strength was observed because if total cellular surface area was most important for precipitating $CaCO_3$ this means there would be no difference in strengths expected for the same total cellular surface area whether it was spread over a relatively high number of smaller cells (i.e., *S. ureae*) or fewer number of larger cells (i.e., *S. pasteurii*).

750 It was the current authors' focus to also apply tests in conditions reflective of a Canadian environment with a relatively novel bacterial isolate (*S. ureae*). Sands treated with *S. ureae* and which underwent short-term flooding (111.67 kPa) or freeze-thaw cycling (93.47 kPa) showed no statistically significant ($p > 0.05$, $n = 3$) strength difference compared to in-lab (135.77 kPa) conditions (Fig. 6). It has been shown that MICP treated sands maintain some porosity in materials (Cheng and Cord-Ruwisch 2012; Chu et al., 2012) and that good strength maintenance in seasonal water saturation and freeze-thaw is possible

755 with porous materials (Cornforth 2005). Further studies may wish to investigate the permeability of hardened sands via *S. ureae* at various levels of $CaCO_3$ precipitation to strike a balance between porosity, peak strength and endurance overtime in weather simulations.

Predictably, it was seen that the acid rain model, reflective of a Northern Ontario rain pH (4.4), eroded the shear strength of sands (Fig. 6) to 35.5 % of originally observed values (Fig. 3). This is a result of the reaction of acid

760 with $CaCO_3$ producing units of $H_2O$, $CO_2$ and salt, known as weathering. A study by Cheng and Cord-Ruwisch (2013) reported similar results with a *Bacillus sphaericus* model. This prompts the idea that a MICP strength model, regardless of the bacteria treatment selected (*S. ureae*, *S. pasteurii*, etc.) for strength enhancement, would require a time-based repair of treated volumes. This realistically limits its geotechnical and economical practicality in the industry. However, it does prompt interest to test the ability of natural buffers, such as limes and sodas, to increase

765 the life-span of MICP induced strength enhancement by reducing acid rain degradation.

**5 Conclusions**

This study has worked to verify that *S. ureae* is a suitable organism to be applied in the soil hardening technology

770 currently being developed via ureolytic MICP. The authors designate it a close ureolytic MICP candidate, in performance, to the well-studied *S. pasteurii* and a superior one to several other *Bacillus* strains. As larger scale simulations are employed, it is strongly encouraged by the authors that further optimization in the treatment procedure, regardless of the MICP organism selected, be undergone including ideal soil buffering to reduce certain climatic effects (i.e., acid rain) and optimum volume porosity in the space to be treated to assure an economical

775 application in industry.

Comment [L136]: L615/616, L616-619, L621, L625/626 : This section rewritten to better describe the role of bacteria as nucleation points for the precipitation of CaCO3 to provide context in the discussion of possible reasons S. ureae treatments did not lead to meaningful strength differences compared to S. pasteurii treatments.

Comment [L137]: Change from 'colony total' or 'cell number' to 'colony abundance' or 'cell abundance'.

Comment [L138]: L635 : Rephrased from 'may prove' to 'whether or not'

Comment [L139]: L639: 'proximal' to 'similar'

Comment [L140]: L646: Change to following to reflect 'retention' of strength.

Comment [L141]: L647: 'remain' to 'maintain'

Comment [L142]: L652: Typographical error ; change from 'reflective or' to 'reflective of'

Comment [L143]: Typographical change ; from 'well studied' to 'well-studied'

**6 Compliance with Ethical Standards**

Funding: The study was funded by NSERC (Discovery grant number 2016-2021 ; Discovery grant number 2015-2020) and the University of Ottawa (UROP grant 2012 ; USRA grant 2014)

780

**7 Competing Interests**

The authors state they have no conflict of interest.

785 **8 Acknowledgements**

The authors would like to acknowledge the University of Ottawa (UROP Grant) and the National Sciences and Engineering Research Council of Canada (NSERC USRA and NSERC Discovery grants to D. Fortin and S. Vanapalli) for financial provisioning in support of this project. Thanks are also given to Mr. Jean Celestin, Mr. Yunlong (Harry) Lui,  Mr. Penghai (Peter) Yin, Dr. Nimal De Silva, Dr. Erika Revesz and Mr. George Mrazek; each providing assistance in shear measurements, microscopy and/or data analysis.

**9 References**

> **Comment [L144]:** Several typographical errors corrected (e.g., line spacing). Additional literature sources also added (see highlighted text).

[revised manuscript text omitted]

      2017, 23 – 25.

---

## Author Response (AR2)

**Author's Response – Editor**

(1)    Author's Response

Dear Dr. Akob,

We appreciate your continued editorial comments and have made the necessary changes in the manuscript below.

Text remains highlighted in yellow to highlight these changes. In addition, comments are provided where necessary.

(2)    Author's Changes in Manuscript

L. 288: Sporosarcina is still spelled wrong.

Spelling corrected.

L. 313-314: correct ddH2O to have subscripts and be characters not numerals for "O"

Subscripts added at L 313-314 and elsewhere in document for 'ddH2O' to '$_{dd}H_2O$'

L. 426: this is still not correct. You are not measuring number of colonies but the OD600. It should be "examination of bacterial abundance in culture" or bacterial biomass or growth. It was changed correctly at L. 462.

Modified to 'bacterial abundance'.

[revised manuscript text omitted]

185 modifications made: -20 °C storage, 1 drop 5 N H$_2$SO$_4$.

Comment [L13]: L166 modification
Description of optical density / cellular density procedure used for all aspects of the study.
If desired the following procedural notes could also be added:

While ddH2O was used as a blank, samples diluted 10-100X had a corresponding TBS sample diluted with medium only in that range.
E.g. Sample 1-10 diluted 20X in TBS (50uL culture volume in 950uL TBS). Then, a sample diluted 20X with medium only was prepared (50mL medium in 950uL TBS). Final OD600 values reported subtracted this value from all sample values measured for a dilution.
For undiluted samples, a sample with medium only was prepared.

Comment [L14]: L166 modifications
Based on previous calibrations performed from recommendations in :

(1) Koch, AL. 1994. "Growth Measurement" IN: Methods for General and Molecular Bacteriology Gerhardt, P et al (ed) American Society for Microbiology, Washington, DC. p. 248-277.
(2) Collins and Lyne's Microbiological Methods" ( Estimating Microbial Numbers – p. 144-155)

Comment [L15]: Typographical error changed

Comment [L16]: L166 modification
A fraction was equal to the volume required to reach OD600 ~0.2 from the equation C1V1 = C2V2 where C1 was the OD600 of the concentrated culture in TBS, and C2 = ~0.2 , V2 = 200mL and V1 = volume to be spun down.

Comment [L17]: Editorial comment
Word change made 'run' to 'incubated'

Comment [L18]: Typographical error changed

Comment [L19]: L139: Change to subscript for OD600.

Comment [L20]: L151 modification
URL provided for the downloadable technical manual of total ammonia measurement by HACH.

Comment [L21]: L151 modification
Sentences hereafter removed.

**2.2.3 Spectrophotometric analysis of $NH_3$-$NH_4^+$**

Samples were thawed and neutralized with 5 N NaOH as described by HACH Inc. (Hach Co. 2015). $NH_3$-$NH_4^+$ measurements were then performed as outlined (HACH Co. 2015) based on an adaptation of the work by Reardon et al. (1966) using a portable DR2700 HACH spectrophotometer. Samples were brought to a measureable range (0.01 to 0.50 mg/L $NH_3$-N) where required. Measurements for appropriate dilutions were made by mass and corrected to volume assuming a density of 1 g/L. Final values were reported as, 'U/mL' where units U = μmol of $NH_3$-$NH_4^+$ produced per minute and mL = mL solution normalized to culture density ($OD_{600}$) starting from t = 1 h.

**Comment [L22]:** L151 modification
Original literature report adapted for total ammonia analysis provided for additional clarity.

**Comment [L23]:** L147 modification
Density assumption stated here only.

**Comment [L24]:** L274 modification
Alternative definition: umol of NH3-NH4+ produced per minute per mL solution, normalized to cellular density (OD600) , where U / mL = umol/min/mL/OD600.

**2.3 Microbial cementation**

**2.3.1 Model sand**

Industrial quality, pure coarse silica sand (Unimin Canada Limited) was examined with the following grain distribution where $D_{10}$, $D_{50}$, $D_{60}$ are 10 %, 50 % and 60 % of the cumulative mass: $D_{10}$ = 0.62 mm, $D_{50}$ = 0.88 mm, $D_{60}$ = 0.96 mm. The uniformity coefficient, $C_u$ was 1.55 indicating a poorly graded (i.e. uniform) sand as designated by the Unified Soil Classification System (USCS) (ASTM, 2017). A poorly graded soil was used as a model due to its undesirable geotechnical characteristics in construction (i.e., settling) and tendency for instability in nature (i.e., liquefaction) (Nakata et al., 2001; Scott, 1991).

**Comment [L25]:** L163: Please see USCS designations for a sand that is 'poorly graded' (i.e. having uniform particle sizes) and 'coarse' grained (i.e. > 50% mass retained on no. 200 sieve with > 50% passing no. 4 sieve) with less than 5% fines.

See ASTM 2011.

A convenient web link:

https://en.wikipedia.org/wiki/Unified_Soil_Classification_System

**2.3.2 Cementation medium (CM) and culture**

[revised manuscript text omitted]

**Comment [L39]:** L220-224
The washing, drying and testing methods are described and made more clear
L230/231 modification
The effect of drying is discussed

**Comment [L40]:** L228-231: Sample mounting details added. No dehydration performed but cleaning with ddH2O and drying by oven was done (as described)/

**Comment [L41]:** L228 : Editorial comment
Replaced 'Visualization' with 'observed'

**Comment [L42]:** Typographical error ; Changed from 'analyzed' to 'analysed'

**Comment [L43]:** L236 modification.
Wording changed to reflect the incubation period.

**Comment [L44]:** Typographical change made

**Comment [L45]:** Editorial comment
Change from 'the trials' and 'withstand' to 'treated silica sand' and 'impacted'.

**2.6 Statistical processing**

All statistical manipulations were performed in Excel (2007). Sample means were reported alongside the standard error of the mean (SE) or standard deviation (SD). Normality of all data sets were confirmed with the Anderson-Darling test ($\alpha = 0.05$). The Student's t-test (unpaired, two-tailed; $\alpha = 0.05$) was utilized to compare sample means of experimental conditions for statistical significance. Prior to each t-test, homogeneity of variances for data sets were determined using a F-test ($\alpha = 0.05$). Where variances were statistically observed as unequal, a Welch's t-test was adapted to test statistical significance between two sample means.

**3 Results**

**3.1 $NH_3$-$NH_4^+$ production**

Among the different bacterial strains considered, *S. pasteurii and S. ureae* were capable of producing the first and second highest levels of $NH_3$-$NH_4^+$, respectively, per unit of time, in both UB-1 (32.50 U/mL ; 29.00 U/mL) and UB-2 (32.76 U/mL ; 30.28 U/mL medium (Fig. 2a, 2b). Isolates of *B. subtilis* (2.91 U/mL), *B. megaterium* (4.87 U/mL) and *L. sphaericus* (5.89 U/mL) displayed a lower peak of $NH_3$-$NH_4^+$ production in both media. When urea in medium moved from the sole source (i.e., UB-2) to one of a number of sources (i.e., UB-1) for nitrogen, $NH_3$-$NH_4^+$ production dropped to near zero values (Fig. 2a, 2b) for *B. subtilis* (0.44 U/mL), *B. megaterium* (0.56 U/mL) and *L. sphaericus* (1.20 U/mL) that were statistically significantly different ($p < 0.05$, $n = 6$) from the final UB-1 values for each species. However, isolates of *S. ureae* and *S. pasteurii* observed no statistically significant difference ($p > 0.05$, $n = 6$) between final values recorded in UB-1 and UB-2 medium. Instead, a rise in production ($t = 0 - 5$ h) followed by a levelling off in value ($t = 6 - 12$ h) was the general trend observed in UB-1 and UB-2 medium (Fig. 2a, 2b).

**Comment [L46]:** L274 : Revised to list production values (U/mL) at end of sentence. L276: Revised to include panel number

**Comment [L47]:** Typographical change made

**Comment [L48]:** Modifications made to clarify that urea was the only source of nitrogen in UB-2 while one of a number of sources for nitrogen in UB-1.

**Comment [L49]:** L279: Revised to include panel number

**Comment [L50]:** 'significant' reworded to 'statistically different'

**Comment [L51]:** 'non significant' reworded to 'no statistically significant difference'

**Comment [L52]:** Typographical change made

**Comment [L53]:** Revised to include panel number

[Figure]

Fig. 2. **(a)** , **(b)** NH₃-NH₄⁺ production (U/mL = umol of $NH_3$-$NH_4^+$ / minute.mL.$OD_{600}$ of culture) ; **(c), (d)** pH ; and **(e), (f)** growth of selected bacteria types in **(a) , (c) , (e)** UB-1 (*No yeast extract [YE]*) and **(b) , (d) , (f)** UB-2 (*10 g/L YE*) nutrient conditions (*SD, n = 6*). YE was a nitrogen source in the growth medium.

Comment [L54]: Additional suggested change based on L274 comment

Comment [L55]: Figure 2:
(1) YE defined as yeast extract and clarified to be a N source in medium.
(2) Change from black to grey scale to better differentiate error bars from data points.
(3) Placement of UB-1 medium panels on left and UB-2 medium panels on right.

**3.2 Examination of bacterial abundance in culture**

All strains showed a decline in growth progression when medium was restricted (i.e., UB-2) to urea as nitrogen and glucose as carbon, sources, respectively (Fig. 2e, 2f). Growth repression was greatest in the cases of *B. subtilis* (-33.9 %), *L. sphaericus* (-26.8 %) and *B. megaterium* (-23.6 %) compared to *S. pasteurii* (-17.8 %) and *S. ureae* (-16.6 %). Additionally, the final $OD_{600}$ (t = 12 h) achieved for all strains in UB-2 medium was decreased compared to UB-1 medium values (t = 12 h) and the difference in value for each strain was found to be statistically significantly different (p < 0.05, n = 6). Growth cessation (i.e. stationary phase) occurred for *S. ureae* and *S. pasteurii* in both conditions but later in UB-1 (t = 11 h) compared to UB-2 (t = 9 – 10 h) medium (Fig. 2e, 2f); they grew logistically in both medium conditions. In general, growth of *L. sphaericus*, *B. subtilis* and *B. megaterium* in UB-2 medium followed a logistic growth curve too. However, in UB-1 medium their growth fit an exponential model, whereby an exponential growth phase was observed from t = 4 – 12 h following a lag phase of growth between t = 0 – 3 h.

**3.3 Changes in pH**

The alkalinity increased with the increase in time for the strains of *S. ureae* and *S. pasteurii* studied, in both UB-1 (8.99, 9.2) and UB-2 (8.74, 8.8) medium. The lowest final pH values were observed in *L. sphaericus* (7.88; 8.16), *B. megaterium* (7.85 ; 7.93) and *B. subtilis* (7.70 ; 7.81) in UB-1 and UB-2 medium, at the end of 12 h (Fig. 2c, 2d). While pH continued to rise for *S. pasteurii* and *S. ureae* in either UB-1 or UB-2 medium, it was constant for *L. sphaericus*, *B. megaterium* and *B. subtilis* after time in UB-1 medium as early as 6 h (*L. sphaericus* and *B. megaterium*) in UB-2 medium. While final pH values for *L. sphaericus*, *B. megaterium* and *B. subtilis* reached higher final (t = 12 h) values in UB-2 medium compared to UB-1, that were found to be statistically significantly different (p < 0.05, n = 6), the opposite was true for *S. pasteurii* and *S. ureae*; values in UB-2 were lower compared to UB-1 and the difference was found to be statistically significantly different for each species (p < 0.05, n = 6). In general, acidity increased with the increase in time for *L. sphaericus*, *B. megaterium* and *B. subtilis* in UB-1 medium. This was also true in UB-2 medium except for *L. sphaericus* which showed an increase in pH over time.

**3.4 Mechanical and biological behaviour in MICP reinforced sands**

Experiments of sand consolidation with triplicate holding vessels (Fig. 1) mixed with *S. ureae* (135.77 kPa) or *S. pasteurii* (135.5kPa) and fed MICP medium (i.e., CM-1) had improvements in their direct shear strength compared to control vessels (15.77 kPa) fed with MICP medium only. In fact, the difference in direct shear strength values for *S. ureae* and *S. pasteurii* compared to control vessels were found to be statistically significantly different (p < 0.05, n = 3). However, the difference in strength between *S. ureae* and *S. pasteurii* were not statistically significantly different (p > 0.05, n = 3). Mixtures of non-ureolytic *B. subtilis* (28.1 kPa) showed no statistically significant difference (p > 0.05, n = 3) in value when compared to the control (Fig. 3). While pre-injection (21.9 x $10^7$ CFU/mL) and post incubation (3.2 x $10^7$ CFU/mL) cell abundance was highest in the case of *B. subtilis*, (Fig. 4) all bacterial isolates showed a decrease in cell abundance when comparing pre-injection to post incubation cell abundance with statistically significant differences (p < 0.05, n = 9). Also, the percentage loss of cell abundance, taken as the difference between post incubation and pre-

**Comment [L56]:** L339: Changed from 'colony total' to 'bacterial abundance'

**Comment [L57]:** Revised to include panel number

**Comment [L58]:** Typographical error correction

**Comment [L59]:** L344-346 correction

**Comment [L60]:** Wording modified to reflect that differences were statistically different and decreased. Also reference section 2.6 for specific tests used, where appropriate.

**Comment [L61]:** Revised to include panel number

**Comment [L62]:** Additional observations noted for growth in medium UB-1 and UB-2

**Comment [L63]:** Revised to include panel number

**Comment [L64]:** L355-357 correction

**Comment [L65]:** Additional details provided on statistical data used (also reference section 2.6 for specific tests used, where appropriate)

**Comment [L66]:** Additional details provided on statistical data used (also reference section 2.6 for specific tests used, where appropriate)

**Comment [L67]:** L368-369: Suggested changes made.

**Comment [L68]:** Additional details provided on statistical data used (also reference section 2.6 for specific tests used, where appropriate)

**Comment [L69]:** Change from 'colony total' to 'cell abundance

410    injection cell abundances divided by the initial pre-injection cell abundance (-77.7 % [*S. ureae*], -75.4 % [*S. pasteurii*], -77.7 % [*B. subtilis*]) were not statistically significantly different (p > 0.05, n = 9) when comparing values between species.  Of note, the medium-only control had no cell growth (CFU/mL) observed before and after incubation.

Comment [L70]: L371-373 corrections

Comment [L71]: Change from 'colony growth' to 'cell growth'

**Comment [L72]:** L386: 'Subtilis' to 'subtilis'

[Figure]

Fig. 3. Direct shear strengths ($\tau$, *kPa*) of treated sands (*SE, n = 3*).

[Figure]

Fig. 4. Microbial viability of treated sands before injection (*black bars*) and after incubation (*gray bars*) (*SD, n = 9*).

**3.5 Microstructure investigation**

450    The precipitation of calcium as $CaCO_3$ via MICP was visualized. Sand granules from approximately the first 1cm of sands treated with MICP solution (i.e., CM-1) combined with *S. ureae* are shown (Fig. 5a, 5b) where crystals arranged in rosette peaks (20 – 40 μm) can be seen across the surface of a sand grain (Fig. 5a, 5b). Rod-shaped structures (40 – 80 μm) can also be visualized, though less commonly, across grain surfaces (Fig. 5a, 5b). Calcium, carbon and oxygen peaks captured by EDS analysis for crystals organized in 'rosette' patterns as well as in rod-shaped structures suggest $CaCO_3$

455    precipitation (Fig. 5c, 5d).

**Comment [L73]:** Revised to include panel number

**Comment [L74]:** Additional corrections made in line with L414/415, L418 and L594 comment

[Figure]

**Comment [L75]:** Arrows rearranged in image (a) to match magnified position in image (b)

[revised manuscript text omitted]

**Comment [L77]:** L509 ; 'chief' changed to the current sentence.

**Comment [L78]:** L512 : Changed to specify that the nutrient medium, with or without yeast extract did not affect the ability for S. ureae to produce NH3-NH4+ from compounds like urea.

**Comment [L79]:** Revised to include panel number

**Comment [L80]:** L518: change of 'mediums' to 'media'

**Comment [L81]:** L521/522 : Additions addressing the limitations of the ammonium/ammonia measurement on the conclusions able to be drawn in the study.

**Comment [L82]:** L509-576: New paragraph created.

**Comment [L83]:** L527 – removed 'see also'

**Comment [L84]:** L532 – removed 'see also'

**Comment [L85]:** L509-576: Sentence reworded to make a new paragraph.

**Comment [L86]:** Revised to include panel number

[revised manuscript text omitted]

**Comment [L87]:** Removed 'see also'

**Comment [L88]:** Typographical error ; change from 'media' to 'medium'

**Comment [L89]:** Typographical error ; addition of 'medium'

**Comment [L90]:** Changed 'co-nitrogen' to 'alternative nitrogen'.

**Comment [L91]:** Revised to include panel number

**Comment [L92]:** Revised to include panel number

**Comment [L93]:** L 549-553 correction

**Comment [L94]:** Revised to include panel number

**Comment [L95]:** Phrase reworded to better clarify other factors contributing to a decreased growth rate overtime.

**Comment [L96]:** L360-L364 : Discussion of why acidity in UB-1 medium would increase over time included.

**Comment [L97]:** Revised to include panel number

**Comment [L98]:** Typographical error; changed 'media' to 'medium'

**Comment [L99]:** L561 : Removed 'returning to S. ureae'. Started new paragraph.

**Comment [L100]:** Revised to include panel number

**Comment [L101]:** Word change; 'high' to 'ureolytic'

**Comment [L102]:** Typographical error; change 'alkalophile' to 'alkaliphile'

**Comment [L103]:** L565; World change, 'Whiffen' to 'Whiffin'

**Comment [L104]:** L566 correction

[revised manuscript text omitted]

**Comment [L105]:** L572-573 ; Included work of Harbottle et al. (2016) as well as Sarmast et al. (2014) to provide context for this work's conclusions in relation to those found by other groups working with S. ureae on MICP.

**Comment [L106]:** L581 : Removed 'only'

**Comment [L107]:** L585 : Changed to better explain improvements in strength of S. ureae treated samples in relation to the control treatments.

**Comment [L108]:** Additional details provided on statistical data used (also reference section 2.6 for specific tests used, where appropriate)

**Comment [L109]:** Typographical error change; 'an' to 'a'

**Comment [L110]:** Typographical error change; 'sp.' to 'species'

**Comment [L111]:** Changed to clarify that evidence supports S. ureae can improve soils by ureolytic MICP, much like S. pasteurii

**Comment [L112]:** L594 : 'Rhombohedra' changed to more qualitative descriptions based on the SEM images provided.

**Comment [L113]:** Revised to include panel number

**Comment [L114]:** Revised to include panel number

**Comment [L115]:** Revised to include panel number

**Comment [L116]:** L594, L600, L600-604: This section is rewritten to better reflect the mineralogy and microscopy techniques used to identify mineral precipitation along MICP treated surfaces.

**Comment [L117]:** Typographical error ; 'media' to 'medium'.

**Comment [L118]:** L597/98: Data not shown but available upon request.

[revised manuscript text omitted]

**Comment [L119]:** L594, L600, L600-604 : This section is added to give context for why SEM analysis was performed.

**Comment [L120]:** Typographical error ; 'analyzed' to 'analysed'

**Comment [L121]:** Change from 'colony total' to 'cell abundance

**Comment [L122]:** Change from 'colony total' to 'cell abundance

**Comment [L123]:** Change from 'colony total' to 'cell abundance

**Comment [L124]:** Changed from 'colony count' to 'serial dilution of the growth medium'

**Comment [L125]:** L606-610 ; This section is reworded to better clarify that differences in the cell counts for S. ureae / S. pasteurii compared to B. subtilis could be due to the dilution medium used for cell counts.

**Comment [L126]:** Typographical error changed ; ',.' to '.,'

**Comment [L127]:** Change from 'colony total' to 'cell abundance

**Comment [L128]:** Change from 'colony total' to 'cell abundance

**Comment [L129]:** L613: Phrase reworded to remove 'non-significant' statement.

**Comment [L130]:** Typographical error change ; change from 'bacterial' to 'bacteria'

**Comment [L131]:** Change from 'cell number' to 'cell abundance

**Comment [L132]:** Typographical error changed : 'modeled' to "modelled"

**Comment [L133]:** Change from 'cell number' to 'cell abundance

**Comment [L134]:** L632: Change from 'colony total' or 'cell number' to 'colony abundance' or 'cell abundance'.

**Comment [L135]:** Change from 'cell number' to 'cell abundance

**Comment [L136]:** Change from 'colony total' or 'cell number' to 'colony abundance' or 'cell abundance'.

sub-optimal spreading mechanism could have hindered strength achievement in some moulds of *S. ureae* treatment where pore plugging by organic matter (i.e., cells) occurred. This in mind, optimization of treatment protocols would help to determine whether or not *S. ureae* is the superior candidate compared to *S. pasteurii* given that it has consistently increased total cell abundance (Fig. 3) to support more nucleation of $CaCO_3$ overtime, in tandem with a $NH_3$-$NH_4^+$production comparable to that of *S. pasteurii*. However, it is important to note that *S. ureae* cells are significantly smaller than cells of *S. pasteurii* (Claus and Fahmy, 1986). Therefore, the total cellular surface area available for nucleation of $CaCO_3$ would be similar for the two species. This provides a possible explanation for why no statistically significant differences in strength was observed because if total cellular surface area was most important for precipitating $CaCO_3$ this means there would be no difference in strengths expected for the same total cellular surface area whether it was spread over a relatively high number of smaller cells (i.e., *S. ureae*) or fewer number of larger cells (i.e., *S. pasteurii*).

It was the current authors' focus to also apply tests in conditions reflective of a Canadian environment with a relatively novel bacterial isolate (*S. ureae*). Sands treated with *S. ureae* and which underwent short-term flooding (111.67 kPa) or freeze-thaw cycling (93.47 kPa) showed no statistically significant (p > 0.05, n = 3) strength difference compared to in-lab (135.77 kPa) conditions (Fig. 6). It has been shown that MICP treated sands maintain some porosity in materials (Cheng and Cord-Ruwisch 2012; Chu et al., 2012) and that good strength maintenance in seasonal water saturation and freeze-thaw is possible with porous materials (Cornforth 2005). Further studies may wish to investigate the permeability of hardened sands via *S. ureae* at various levels of $CaCO_3$ precipitation to strike a balance between porosity, peak strength and endurance overtime in weather simulations.

Predictably, it was seen that the acid rain model, reflective of a Northern Ontario rain pH (4.4), eroded the shear strength of sands (Fig. 6) to 35.5 % of originally observed values (Fig. 3). This is a result of the reaction of acid with $CaCO_3$ producing units of $H_2O$, $CO_2$ and salt, known as weathering. A study by Cheng and Cord-Ruwisch (2013) reported similar results with a *Bacillus sphaericus* model. This prompts the idea that a MICP strength model, regardless of the bacteria treatment selected (*S. ureae*, *S. pasteurii*, etc.) for strength enhancement, would require a time-based repair of treated volumes. This realistically limits its geotechnical and economical practicality in the industry. However, it does prompt interest to test the ability of natural buffers, such as limes and sodas, to increase the life-span of MICP induced strength enhancement by reducing acid rain degradation.

**5 Conclusions**

This study has worked to verify that *S. ureae* is a suitable organism to be applied in the soil hardening technology currently being developed via ureolytic MICP. The authors designate it a close ureolytic MICP candidate, in performance, to the well-studied *S. pasteurii* and a superior one to several other *Bacillus* strains. As larger scale simulations are employed, it is strongly encouraged by the authors that further optimization in the treatment procedure, regardless of the MICP organism selected, be undergone including ideal soil buffering to reduce certain climatic effects (i.e., acid rain) and optimum volume porosity in the space to be treated to assure an economical application in industry.

**Comment [L137]:** L615/616, L616-619, L621, L625/626 : This section rewritten to better describe the role of bacteria as nucleation points for the precipitation of CaCO3 to provide context in the discussion of possible reasons S. ureae treatments did not lead to meaningful strength differences compared to S. pasteurii treatments.

**Comment [L138]:** Change from 'colony total' or 'cell number' to 'colony abundance' or 'cell abundance'.

**Comment [L139]:** L635 : Rephrased from 'may prove' to 'whether or not'

**Comment [L140]:** L639: 'proximal' to 'similar'

**Comment [L141]:** L646: Change to following to reflect 'retention' of strength.

**Comment [L142]:** L647: 'remain' to 'maintain'

**Comment [L143]:** L652: Typographical error ; change from 'reflective or' to 'reflective of'

**Comment [L144]:** Typographical change ; from 'well studied' to 'well-studied'

**6 Compliance with Ethical Standards**

Funding: The study was funded by NSERC (Discovery grant number 2016-2021 ; Discovery grant number 2015-2020) and the University of Ottawa (UROP grant 2012 ; USRA grant 2014)

725

**7 Competing Interests**

The authors state they have no conflict of interest.

730 ## 8 Acknowledgements

The authors would like to acknowledge the University of Ottawa (UROP Grant) and the National Sciences and Engineering Research Council of Canada (NSERC USRA and NSERC Discovery grants to D. Fortin and S. Vanapalli) for financial provisioning in support of this project. Thanks are also given to Mr. Jean Celestin, Mr.
735 Yunlong (Harry) Lui, Mr. Penghai (Peter) Yin, Dr. Nimal De Silva, Dr. Erika Revesz and Mr. George Mrazek; each providing assistance in shear measurements, microscopy and/or data analysis.

**9 References**

> **Comment [L145]:** Several typographical errors corrected (e.g., line spacing). Additional literature sources also added (see highlighted text).

[revised manuscript text omitted]